# Towards a bottom-up understanding of antimicrobial use and resistance on the farm: A knowledge, attitudes, and practices survey across livestock systems in five African countries

Mark A. Caudell[1]*, Alejandro Dorado-Garcia[2], Suzanne Eckford[2], Chris Creese[2], Denis K. Byarugaba[3], Kofi Afakye[4], Tamara Chansa-Kabali[5], Folorunso O. Fasina[6], Emmanuel Kabali[7], Stella Kiambi[1], Tabitha Kimani[1], Geoffrey Mainda[8], Peter E. Mangesho[9], Francis Chimpangu[10‡], Kululeko Dube[7‡], Bashiru Boi Kikimoto[11‡], Eric Koka[12‡], Tendai Mugara[7‡], Bachana Rubegwa[6‡], Samuel Swiswa[13‡]

1 Food and Agriculture Organization of the United Nations, Nairobi, Kenya, 2 Food and Agriculture Organization of the United Nations, Rome, Italy, 3 College of Veterinary Medicine, Makerere University, Kampala, Uganda, 4 Food and Agriculture Organization of the United Nations, Accra, Ghana, 5 Department of Psychology, University of Zambia, Lusaka, Zambia, 6 Food and Agriculture Organization of the United Nations, Dar es Salaam, Tanzania, 7 Food and Agriculture Organization of the United Nations, Harare, Zimbabwe, 8 Department of Veterinary Services, Ministry of Fisheries and Livestock, Lusaka, Zambia, 9 National Institute for Medical Research, Amani Medical Research Centre, Muheza, Tanzania, 10 Food and Agriculture Organization of the United Nations, Lusaka, Zambia, 11 Public Health & Food Safety Unit, Veterinary Service, Accra, Ghana, 12 Department of Sociology and Anthropology, University of Cape Coast, Cape Coast, Ghana, 13 Division of Veterinary Services, Department of Livestock and Veterinary Services, Harare, Zimbabwe

☯ These authors contributed equally to this work.
‡ These authors also contributed equally to this work.
* mcaudell@wsu.edu

## Abstract

The nutritional and economic potentials of livestock systems are compromised by the emergence and spread of antimicrobial resistance. A major driver of resistance is the misuse and abuse of antimicrobial drugs. The likelihood of misuse may be elevated in low- and middle-income countries where limited professional veterinary services and inadequately controlled access to drugs are assumed to promote non-prudent practices (e.g., self-administration of drugs). The extent of these practices, as well as the knowledge and attitudes motivating them, are largely unknown within most agricultural communities in low- and middle-income countries. The main objective of this study was to document dimensions of knowledge, attitudes and practices related to antimicrobial use and antimicrobial resistance in livestock systems and identify the livelihood factors associated with these dimensions. A mixed-methods ethnographic approach was used to survey households keeping layers in Ghana (N = 110) and Kenya (N = 76), pastoralists keeping cattle, sheep, and goats in Tanzania (N = 195), and broiler farmers in Zambia (N = 198), and Zimbabwe (N = 298). Across countries, we find that it is individuals who live or work at the farm who draw upon their knowledge and experiences to make decisions regarding antimicrobial use and related practices. Input from

**Data Availability Statement:** All relevant data are within the manuscript and its Supporting Information files.

**Funding:** This research was funded by a grant from the Fleming Fund of the United Kingdom (https://www.flemingfund.org/) to the Food and Agriculture Organization of the United Nations (GCP/GLO/710/UK). The funder had no role in study design, data collection and analysis, decision to publish, or preparation of the manuscript.

**Competing interests:** The authors have declared that no competing interests exist.

animal health professionals is rare and antimicrobials are sourced at local, privately owned agrovet drug shops. We also find that knowledge, attitudes, and particularly practices significantly varied across countries, with poultry farmers holding more knowledge, desirable attitudes, and prudent practices compared to pastoralist households. Multivariate models showed that variation in knowledge, attitudes and practices is related to several factors, including gender, disease dynamics on the farm, and source of animal health information. Study results emphasize that interventions to limit antimicrobial resistance should be founded upon a bottom-up understanding of antimicrobial use at the farm-level given limited input from animal health professionals and under-resourced regulatory capacities within most low- and middle-income countries. Establishing this bottom-up understanding across cultures and production systems will inform the development and implementation of the behavioral change interventions to combat antimicrobial resistance globally.

## Introduction

Antimicrobial drugs, such as antibiotics, are essential to protect animal health in livestock production systems but their misuse and/or abuse selects for the emergence, transmission, and persistence of antimicrobial resistance (AMR), the phenomenon where microbes ['germs'] acquire the ability to tolerate one or more drugs we rely on to treat microbial infections [1–3]. The emergence of AMR is resulting in longer and/or more cycles of treatment, as well as therapeutic failures threatening animal welfare, food security, and public health worldwide. AMR in animals may affect the prevalence of infectious disease in people as resistant microorganisms (e.g., bacteria, viruses, parasites) can be transmitted across the human-animal interface through a diverse set of pathways, including consumption of animal products, direct contact and sharing of water sources [2,4]. This potential for widespread transmission and the ramifications of treatment failures are significant, especially considering that nine of the fourteen classes of drugs labelled 'critically important' in public health are also used in livestock systems [5]. Antimicrobial use in the agricultural sector is projected to increase by 67% by the year 2030, potentially further compromising the effectiveness of medicines in both animal and human health [6]. The interconnectedness of antimicrobial use in agriculture and public health makes AMR the "quintessential One Health issue" of our time [7] requiring collaboration across disciplines, sectors (public and private) and scales (locally, globally) to better manage AMR in people, animals, and the environment [8].

The One Health challenges of AMR are particularly striking in low- and middle-income countries (LMICs) given disproportionately high burdens of infectious disease, alongside livelihoods and living conditions promoting frequent interactions between people and livestock. Genotypic studies in LMICs provide evidence for the transmission of AMR across people, animals and the environment. In Tanzania, for example, genotypic similarity has been documented between resistant enteric bacteria in people, animals (both livestock and wildlife) and the environment (i.e., waters sources) [9,10] Similar patterns were found in *Salmonella* isolates from people and animals in Uganda [11]. In contrast, genotypic studies from high-income countries have largely shown distinguishable epidemics of AMR in livestock and the general population [12–15]. These patterns are consistent with limited contact between the general population and livestock and with the development of health, sanitation, and regulatory infrastructures that limit transmission events [12–14,16,17]. While a study conducted in

Netherlands did document overlap between livestock (pigs) and people, the overlap was dependent on intensity of contact and was higher in farming communities [18]. These studies demonstrate that intervention efforts to limit AMR must be tailored to regional and local realities. Therefore, research investigating the particular drivers of AMR and antimicrobial misuse at regional, national, and local scales is urgently needed.

Within most LMICs, the combined realities of underfunded veterinary healthcare systems and limited regulatory capacities constrain efforts to promote prudent antimicrobial use and control AMR in the agricultural sector [19–22]. Increasingly, these efforts will be guided by multisectoral National Action Plans whose activities are supported by the Tripartite Collaboration on AMR, consisting of the Food and Agriculture Organization (FAO), the World Organisation for Animal Health (OIE), and the World Health Organization (WHO). National Action Plans set a series of goals to improve awareness on AMR and related threats, develop capacity for surveillance and monitoring of antimicrobial use and AMR, strengthen governance, and promote antimicrobial stewardship within the public and animal health sectors [23,24]. However, successful implementation of these plans within the livestock sector is limited by poor access to animal health professionals (see Fig 1). In the five African countries surveyed in this study, for example, the ratio of veterinarians to livestock is about 20 times lower than that of high-income countries of Denmark, France, Spain and the USA (data obtained from WAHIS and FAO STAT). Accessibility issues are common across resource-limited countries given private sector healthcare services have inadequately compensated for reductions in public services forced by Structural Adjustment Programs in the 1980's [25,26]. Further limiting achievement of National Action Plans are under resourced regulatory authorities (e.g., national medical authorities, food safety departments). This renders national regulations, such as laws mandating prescriptions for antimicrobials or regulations on antimicrobial residues, difficult to enforce and this (i.e., inability to regulate residues) can have a significant impact on trade [22]

Top-down implementation and enforcement of National Action Plans within LMICs carries a benefit of speed and scale (when resources are available). However, 'bottom-up' approaches to behavioral change—involving stakeholders as early as possible in the process—tend to yield more sustainable change. Thus, countries may benefit most from an integrative approach, and these 'bottom-up' interventions will depend on an understanding of the socio-cultural, economic, and historical factors that motivate antimicrobial use and related practices (e.g., observance of withdrawal) in livestock systems. Unfortunately, there is currently little information on antimicrobial use practices and motivating factors in livestock systems for most LMICs [2,21,27]. Available studies generally find that farmers administer antimicrobials themselves, and mostly without prescriptions or using input from animal health professionals, as well as engaging in other non-prudent practices, such as violating antimicrobial withdrawal periods [28–37], the period of time before slaughter when treatments for the animal must cease in order to effectively eliminate them from the animal's system. However, antimicrobial use patterns vary across subsistence types (e.g., pastoralists, highland farmers), farm size (e.g., small-scale versus commercial) and location (urban versus rural). In a study of antimicrobial use across three groups in northern Tanzania, for example, Caudell et. al [28] found that lay administration (i.e., use by nonprofessionals) is highest among Maasai pastoralists (>90%) and Arusha agropastoralists (>70%) and lowest among Chagga highland farmers (<5%). Widespread lay administration has also been documented in another study of Tanzanian Maasai [38], in Kenyan Maasai [32,39], in cattle keepers in Eastern Province, Zambia [40] and poultry farmers in Ghana [29,33], Kenya [41], and Tanzania [42].

Studies examining antimicrobial treatment patterns, while mostly reliant on self-reported data, have documented other deviations from recommended practices. Across countries, antimicrobials are almost always purchased without prescriptions at 'agrovets' (shops selling

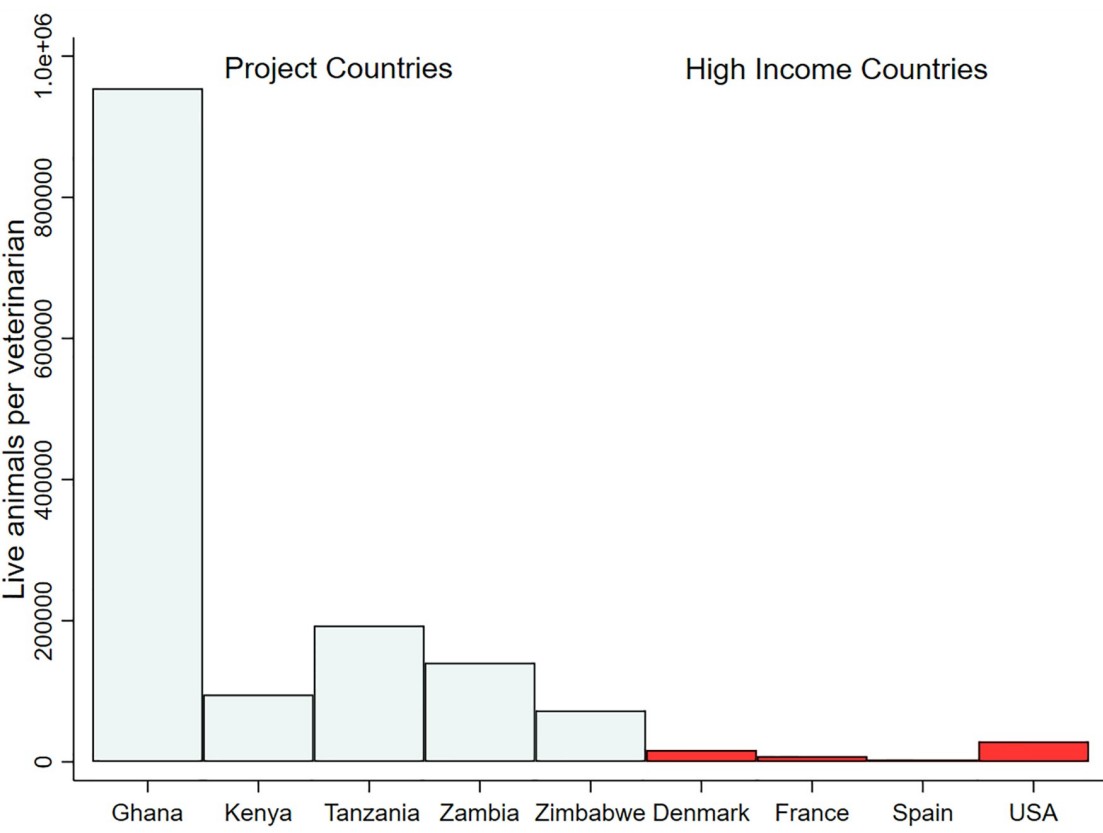

**Fig 1. Live animals per veterinarian for project countries compared with high income countries.** Data obtained from WAHIS and FAO-STAT.

animal health products) and to a lesser extent at open markets and feed distributors [28–30,32–37]. In terms of treatment, a study of poultry farmers in eastern Ghana found that only 63% of farmers completed recommended antibiotic course durations [33]. When administering oxytetracyclines, Maasai pastoralists were observed to give only a single injection when the normal treatment calls for multiple injections [32]. In eastern Zambia, cattle-keepers were also found to underdose [40]. While previous studies of farmers in the five African countries for this project reported antimicrobial use primarily for therapeutic purposes [28,31] preventative use was more frequent in poultry systems [30,34]. Using antimicrobials for 'growth promotion' (faster animal fattening) appears to be limited in the project countries [31] but few households reported compliance with antimicrobial withdrawal periods, noting consumption and sale of products (meat, milk, eggs) from animals receiving treatments during the withdrawal period [28–31,38,43].

While studies examining antimicrobial use and AMR relevant practices are available, more comprehensive studies of the knowledge, attitudes, and practices (KAP) associated with antimicrobial use in particular regions (and sectors) is critically needed to identify 'risky' behaviors (and factors contributing to them) as potential targets for intervention. We therefore conducted KAP surveys across 887 farms in five African countries to look for potential commonalities as a starting point for the development of an intervention program for practical and sustainable changes in behavior. The systems surveyed included pastoralist communities in Tanzania, large-scale and intensive commercial poultry farmers in Ghana (layers) and small-

scale commercial poultry farmers in Kenya (layers), Zambia (broilers), and Zimbabwe (broilers). The primary objective of this paper is to identify the distribution and correlates of knowledge, attitudes, and practices regarding antimicrobial use and AMR in these communities. Knowledge in this context is defined as level of understanding about AMR, antibiotics, drug residues, dosage regimens, and vaccines; attitudes are determined as level of sensitivity to the risks of antimicrobial use and appropriate use of antibiotics and alternatives (vaccines); and practices as level of implementation of actions promoting prudent use that help to prevent the emergence and spread of AMR.

## Materials and methods

### Selection of study locations and production systems

Countries included in this study are part of the UN FAO project "Engaging the food and agriculture sectors in sub-Saharan Africa and South and South-east Asia in the global efforts to combat antimicrobial resistance using a One Health approach" (Project GCP/GLO/710/UK). This project is funded by the government of the United Kingdom of Great Britain and Northern Ireland (Fleming Fund) and provides support to countries in sub-Saharan Africa and south and southeast Asia. Twelve countries were selected for inclusion on the basis of the priority regions/countries of the UK overseas development programmes and taking into account the political and policy environment in the country, the extent of high level engagement in country for addressing AMR in the One Health context and the potential to create country hubs that could support efforts by other countries to address AMR. The project objectives encompass support for development and implementation of One Health National Action Plans to combat AMR, including approaches to improve stakeholder behaviors in regard to antimicrobial use, with specific activities tailored according to country set priorities. Within Africa, five countries (Ghana, Kenya, Tanzania, Zambia and Zimbabwe) were selected to participate in this regionally harmonized KAP survey approach.

Within countries the sampling was not representative of any given geographic region. Sampled localities as well as targeted production systems were identified by members of national FAO teams and government officials. The FAO national teams prioritized where possible the geographical units (e.g., districts) where the targeted production system had the largest numbers of households engaged in the system across the country. Targeted production systems were chosen based several criteria. Poultry systems were selected for four of the five countries given that: 1) these systems were projected to grow at some of the fastest rates both in terms of contribution to GDP and number of households participating in these systems, and, 2) the poultry industry is the highest consumer of antimicrobials. For Tanzania, the FAO team chose to focus on pastoralists as there is limited research of antimicrobial use within these systems and to contrast these systems with poultry systems. For more detailed reasons behind within-country sampling and production system selection see S1 Text. Fig 2 provides a map of the countries surveyed, with approximate survey locations within each country, the production system surveyed, and corresponding sample size for each country.

### KAP survey development and deployment

The project was developed and implemented by an interdisciplinary research team comprised of animal health experts, epidemiologists and social scientists from FAO, national ministries of agriculture and livestock, and within-country academic institutions. The ethnographic approach employed was a modified version of an exploratory cross-sectional survey design [44], which uses qualitative data to inform quantitative surveys and confirm quantitative measures. Focus group discussions and key informant interviews were conducted with

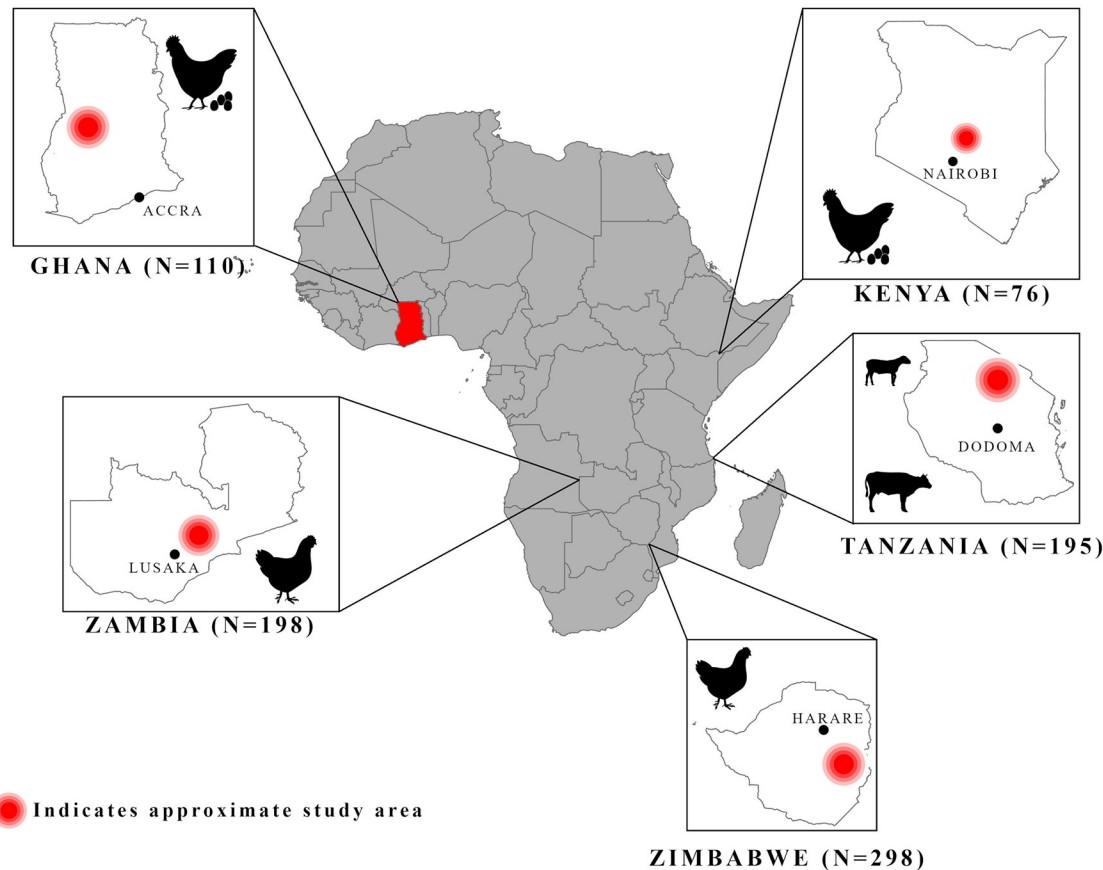

**Fig 2. Project map.** Surveyed countries are highlighted in red. Insets of country maps include country capitals, approximate study area (denoted in red circle), icons for production system surveyed (broiler, layer, pastoralist), and associated sample sizes below. See map legend for description of map markers. Maps were created using ArcGIS software by Esri. The base map is sourced from Esri and modified in ArGIS Pro. "Light Gray Canvas" [basemap] https://www.arcgis.com/home/item.html?id= ee8678f599f64ec0a8ffbfd5c429c896. May 13th, 2019.

stakeholders influencing patterns of antimicrobial use, including farm owners, farm managers, farm workers, animal health professionals (e.g., public and private veterinarians, community animal health workers, para-veterinarians), owners and employees of agrovet shops selling antimicrobials and other livestock products (e.g., feed, disinfectant, vitamins) and feed producers and distributors in poultry systems. When possible, we attempted to interview a minimum of three to a maximum of six groups within each stakeholder category. Three to six groups were targeted given that 90% of the information within a research topic is normally discoverable with this range (i.e., the saturation point of a topic) [45].

Qualitative interviews with stakeholders were concentrated around major themes associated with antimicrobial use and AMR including farm management practices, local disease histories, health-seeking behaviors and health infrastructures, and governance, regulations, policies, and enforcement related to use and AMR. Thematic analysis of qualitative data from all stakeholders within a country was used to initially develop and refine a KAP survey instrument of over 200 items that covered the themes emerging from the qualitative work along with basic demographic and economic questions. All questions concerning disease histories, antimicrobial use, health-seeking practices were asked in the context of the targeted production system. For example, in broiler systems, enumerators would ask "What diseases do *your*

*broilers* commonly get?". In addition, each KAP survey was tailored to a the production and country context (including local language). However, questions covering antimicrobial use and AMR knowledge, attitudes, and practices were harmonized across countries and questions concerning animal health, demographics, and health advice seeking practices were often the same. See S1 File for an example of a KAP survey administered in Zambia.

KAP questionnaires were administered by local enumerators using Kobo Collect®, an Android-based application loaded onto tablets. Enumerators were instructed to survey household heads and/or spouses of household heads where possible. Enumerators received three to four days of training and assisted in refining the survey as most had an animal health background and worked in areas near the study communities. Surveys were piloted prior to administration to ensure question clarity, conduct further refinement, and verify survey times. Pilots were conducted in farms not included in the study and at locations of enumerator training (Nairobi, Kenya; Harare, Zimbabwe, Kabwe, Lusaka, and Longido, Tanzania). Across the five project countries, surveys required between 45 minutes and 1 hour and 30 minutes to complete. All informants provided consent. Prior to requesting for consent, an information sheet containing a detailed narrative of the study and its aims was provided to potential participants who could read and was read out to those who could not. Participants were informed of the research purposes including the benefits and risks of participation. The respondents were assured of their right to withdraw from study participation at any point, and necessary precautions were made to insure and maintain confidentiality, anonymity and voluntarism throughout the study. A written informed consent was sought from all study participants who could write. For those who could not write a thumbprint signature was requested.

## Sampling

Several sampling strategies were used across project countries given differences in availability of census records, reachability of households, financial resources, and time constraints. In most countries, a purposive sampling approach (i.e., non-probability sampling based upon researcher judgment) was used given a lack of information on the households that participated in a particular production system (e.g., the number of households in a community that kept broilers). Due to this lack of information, (e.g. no or outdated national records or farm registration process), the research teams had to rely on the judgements of locals/respondent to identify those within the community who were engaged, or had been, in a particular production system. In Kenya and Zambia, enumerators developed lists of households keeping layers or broilers, respectively, by walking house-to-house and/or consulting with local leaders and veterinary officers. These sample frames were then consulted to schedule interviews. In Ghana and Zimbabwe, existing lists of farmers were used alongside a 'snowball sampling' approach where farmers identified other poultry farmers within the community. For Tanzania, two selection strategies were used including visiting Maasai households and market places. Importantly, given variability in sampling and associated resources across countries, we did not achieve the sample size of households as determined by power calculations (approx. 400 hhs) to develop representative samples for our targeted geographical areas. As such, statements such as "Kenyan farmers" should be considered as "Kenyan farmers surveyed in this study".

## Ethical approvals

Ethical approvals were received in each country. For Ghana, the study was approved by the Ministry of Health Ethics Review Committee (ID No. 014/10/18). For Kenya, the study was approved by the AMREF Health Africa Ethics and Scientific Review Committee (AMREF-ESRC P551/2018) and the Institutional Animal Care and Use Committee of KALRO-Veterinary

Science Research Institute (KALRO-VSRI/IACUC016/28092018). For Tanzania, the study was reviewed and approved by the Medical Research Coordinating Committee of the National Institute for Medical Research (NIMR) in Tanzania and certificate clearance no. NIMR/HQ/R.8a/Vol.IX/2926 was issued. For Zambia, the study was reviewed and approved by the ERES Converge Ethic Committee, Ref: No 2018-Nov-020 was issued. For Zimbabwe, approval was obtained from the Agriculture Research Council (ARC) of the Department of Veterinary Services & Crop and Livestock (Reference Number: 008/2018).

## Variable description

Linear scales of knowledge, attitudes and practices are defined in Table 1. Scales were generated from the variables included in Table 2 and used as outcomes in ordinal least squares regression models. Scales were developed by recoding variable answers to binary with 1 representing sufficient knowledge to understand antimicrobials and AMR, desirable attitudes to address AMR, and appropriate practices for controlling AMR and 0 representing insufficient knowledge, undesirable attitudes and inappropriate practices. For attitudes and practices scales, responses of "indifferent" and "sometimes" were coded undesirable and inappropriate, respectively. Responses were then summed for each participant and divided by the total number of items within the category to arrive at a percentage of correct answers. For example, if a respondent reported observing 6 of the 8 appropriate practices, they received a 75% (6/8) prudent practice score out of a possible 100%. One-Way ANOVA analysis were used to assess significant differences in KAP scores across countries. Tukey-Kramer comparisons, which conduct pairwise testing of means in One-Way ANOVAs with sample sizes that are unequal [46] were used to assess which country comparisons were significant. Pearson correlations were used to calculate the associations between KAP scores across and within countries.

## Modeling approach

Multivariate ordinal least squares regression was used to assess the factors associated with KAP scales. These regression models were used to assess three domains important for intervention design, including demographics, on-farm dynamics, and health seeking practices (see description of domains below). For each domain, models were specified by production system, including models for poultry systems and models for pastoralist systems. The decision to examine these production systems separately was due to two reasons. First, given the different practices characterizing these systems, some variables were not available for a combined analysis (e.g., differences in biosecurity practices across poultry and pastoralist systems). Second, earlier pooled analysis that included all production systems indicated the presence of significant interaction effects between Tanzania (pastoralist system) and potential correlates of KAP related to antimicrobial use and AMR. Importantly, interaction effects were not significant for either broiler or layer countries and so systems were combined for analysis. In poultry models, however, we still control for country-level effects by entering country as a dummy variable. While our interest is in assessing how different domains impact knowledge, attitudes,

**Table 1. Definition of KAP scales.**

| Scale | Meaning |
|---|---|
| Knowledge | level of understanding of AMR, antimicrobials, residues, dosage regimes, and vaccines |
| Attitudes | level of awareness towards appropriate use of antimicrobials and alternatives (vaccines), including sensitiveness to risks from antimicrobial use |
| Practice | level of implementation of practices that prevent AMR and promote prudent antimicrobial use |

**Table 2. Summary of survey questions and resulting variables included in knowledge, attitudes, and practices scale.** For specific wording of questions see S1 File.

| Original variables | Recoded values |
|---|---|
| **Knowledge** (Yes = 1 No = 0) | |
| Able to explain antimicrobial resistance | 1 = 1; 0 = 0 |
| Able to explain what antibiotics are/do | 1 = 1; 0 = 0 |
| Able to explain what antibiotic residues are/do | 1 = 1; 0 = 0 |
| Able to explain dosage/treatment of commonly used antibiotic | 1 = 1; 0 = 0 |
| Able to explain what vaccines are/do | 1 = 1; 0 = 0 |
| **Attitudes** (disagree = 1, neutral/indifferent = 2, agree = 3) | |
| If medicines are given too often then they might stop working | 1&2 = 0; 3 = 1 |
| Giving animals that are not sick antimicrobials will prevent them from becoming sick in the future | 1 = 1; 2&3 = 0 |
| Giving animals that are not sick antimicrobials can help them grow bigger, faster, fatter, boost egg production/size | 1 = 1; 2&3 = 0 |
| It is important to get consultation from a veterinarian before giving antimicrobials to the animals | 1&2 = 0; 3 = 1 |
| Using vaccines can reduce use of antibiotics | 1&2 = 0; 3 = 1 |
| After using antibiotics on an animal, you should wait sometime before using the products from it, such as bird meat/eggs/milk | 1&2 = 0; 3 = 1 |
| **Practices** (never/rarely = 0, sometimes = 1, almost always = 2) | |
| Give antimicrobials when get you day-old chicks/new calves/smallstock | 0 = 1; 1&2 = 0 |
| Give antimicrobials to all animals when one is sick | 0 = 1; 1&2 = 0 |
| Give animals a larger dose than recommended | 0 = 1; 1&2 = 0 |
| Give birds a smaller dose than the recommended dose? | 0 = 1; 1&2 = 0 |
| Stop using antimicrobials before the full dose because the animal has improved | 0 = 1; 1&2 = 0 |
| Use expired medicines | 0 = 1; 1&2 = 0 |
| Have a prescription when purchasing antibiotics | 0&1 = 0; 2 = 1 |
| Observe withdrawal from antimicrobials | 0&1 = 0; 2 = 1 |

and practices related to antimicrobial use and AMR, we also recognize the potential importance of omitted-variable bias. To assess the potential impact of this bias we also specified models (one for poultry and one for pastoralism) that combined variables from all domains together (demographics, on-farm dynamics, and health seeking practices). We have included these models in Tables 3-Tables 4 in S1 Text. If results from the full models fundamentally differ from domain-specific models we highlight these differences in the discussion.

Variables included in the model for demographics were age, gender, education and years engaged in the targeted production system (e.g., number of years a respondent kept broilers) (see Table 3). The model for on-farm dynamics included variables related to farm size, biosecurity, disease histories, antimicrobial use patterns, trainings on farm management and record-keeping. The model for health seeking model included variables on frequency which people sought advice from various stakeholders (e.g., friends, veterinarians, agrovets).

Regression diagnostics were assessed for all models. Influential data points were examined through calculating Cook's D, a measure that is calculated for each data point that shows the influence of the point on the fitted response values [47]. Models were run without observations associated with values exceeding 4/N (4/857 = 0.005). Excluding observations above this threshold did not impact model interpretation. Multicollinearity was assessed calculating variance inflation factors (VIFs). Variance inflation factors for variables of interest were less than 2, below the recommended cut-off of 5[48]. Variance inflation factors for some country-control variables (VIF≈6) were above the cut-off. However, we do not interpret the coefficients of

**Table 3. Definitions of variables and variable types included in KAP studies.**

| Variable | Definition |
|---|---|
| Gender | Female = 1 Male = 0. |
| Age | Respondent age. Continuous |
| Education level | Education level was none primary, secondary, and tertiary and above. Tertiary and above indicates any additional education after secondary school, including certificates, diplomas, bachelors, masters, and PhDs. Education levels were dummy coded and entered into the models with "none" being the omitted variable. |
| Farm scale | Total number of animals kept on the farm standardized at the country level. Continuous |
| Treatment failure | (Yes = 1, No = 0) Whether a respondent has noticed an increase in treatment failure with antimicrobials on their farm. |
| Disease level | Percentage representing the number of diseases reported by the household as common divided by the total number of diseases listed across a community Continuous |
| AMs used per month | The number of antimicrobial products a person recorded using in the last month within the targeted system. Continuous |
| Number AM medicines | The number of antimicrobials reported by the respondent as commonly used. Continuous |
| Keep records | (Yes = 1, No = 0) farmer reported keeping records on one or more of medicines used, mortality statistics, purchases and sales. |
| Keeping time | The number of years a person had been engaged in the target production system. Continuous |
| Training | (Yes = 1, No = 0) included training on animal health, biosecurity, production, and marketing. |
| Advice variables. | Advice variables indicating whether the source never/rarely provided advice, sometimes, and almost always. Advice levels were dummy coded and entered into the models with "none" being the omitted variable. |

control variables and these controls were not collinear with variables of interest as indicated by low VIF values, ensuring the performance of the controls was not impaired [49]. Normality of residuals was tested with the Shapiro-Wilk Test [50], kernel density, and standardized normal probability plots. Results from all tests indicted only slight deviations from normality. The homoscedasticity assumption (i.e., homogeneity of variance of residuals) were tested using residuals versus fitted (predicted) plots [51]. There was evidence of minor heteroscedasticity in the demographic-factors model so a Huber-White sandwich estimator was used to provide robust estimates [52].

# Results

## Qualitative results: A brief overview

A more comprehensive analysis of the qualitative data is currently in progress, but we provide a preliminary analysis of major themes emerging from discussions with farmers, animal health professionals, and agrovets across the project countries. In general, we found that farmers were often aware of the negative associations between biosecurity and disease and antimicrobial use but economic considerations prevented them from investing in biosecurity. A layer farmer in Kenya, for example, explained "we know we should have footbaths, but we don't have them. Most monies go to feed and medicines, so we don't have a lot of money to concentrate on other side costs like footbaths and disinfectants–they are costly". Another common challenge listed by farmers were the difficulties of accessing animal health professionals. As a broiler farmer in Zimbabwe said, "the VAs [veterinary assistants] are not giving us the desired response no matter how urgent the matter is, we do not know whether they are too busy, or they concentrate more on commercial farmers". In Ghana, a layer farmer explained that "The veterinary officer will come but because he has no equipment, he will be doing guess work"

Farmer perspectives were consistent with qualitative interviews among animal health professionals, whose major complaint was that they lacked the resources to provide proper services. "Our annual budget", explained a veterinarian, "is around 570 USD per year. That's less than 50 USD a month, so we have limited money for fuel and equipment. Further, it is difficult to have trainings due to lack of resources". Another common theme, again consistent with farmer interviews, was that animal health professionals pointed to poor biosecurity practices as the main driver of antimicrobial use. Professionals across the project countries all agreed that they were perceived as the "last resorts" for animal health, with farmers only calling them after several treatment strategies have failed. In support, our qualitative interviews with agrovet employees showed they were often one of the first sources of health advice sought by farmers. Discussions with agrovets indicated that, across countries, most had knowledge of AMR and recognized the importance of getting drug prescriptions from the farmers. Although acknowledging this importance, most agrovet dealers confessed they usually sell antibiotics without prescriptions, basing drug decisions off symptom descriptions or specific requests of the farmer. These practices continue because government regulatory authorities meant to enforce these laws are a "rare sight". "I have been keeping this shop for 15 years", said one agrovet, "and I have been visited by [the agency], two times to ask about my drugs".

## Descriptive results

In our sample, Ghana farmers kept the largest flocks with a median flock size of 4054 birds ($Q_1$ = 2000, $Q_3$ = 9000) while Kenya farms had a median of 700 birds ($Q_1$ = 300, $Q_3$ = 1150). Broiler systems were smaller than layer systems with a median of 105 birds in Zambia ($Q_1$ = 0, $Q_3$ = 250) and 100 birds in Zimbabwe ($Q_1$ = 25, $Q_3$ = 150). Flock mortalities, calculated as percentage of average flock size, were highest in layer systems in Kenya ($\approx$16%) and Ghana ($\approx$14%) and slightly lower in broiler systems in Zambia ($\approx$10%) and Zimbabwe ($\approx$8%). In terms of biosecurity, a minority of farms had footbaths with Zambia farmers reporting the highest rate of footbath ownership (48%) followed by Zimbabwe (22%), Ghana (6%) and Kenya (3%). Around 70% of farms owned boots for the poultry houses except for farmers in Zimbabwe where only 32% reported owning boots. Ninety nine percent of farmers in Ghana reported keeping farm records, followed by $\approx$80% of Zambian farmers, $\approx$70% Zimbabwean farmers, and $\approx$60% of Kenyan farmers. The most common records kept on the farm were sales records and flock mortalities. Medicine costs, including antimicrobials and vaccines, were the highest in Ghana (0.65 USD per bird), followed by Zimbabwe (0.31 USD), Zambia (0.23 USD) and Kenya (0.21 USD). The top three self-reported diseases in layers were Coccidiosis ($\approx$63%), Chronic Respiratory Disease (CRD) (76%) and Newcastle Disease (39%). For broilers, the top three self-reported diseases were Coccidiosis ($\approx$43%), Chronic Respiratory Disease ($\approx$32%), and Infectious Bronchitis ($\approx$30%).

Although of comparable age ($\approx$45 years), Maasai pastoralists were distinguished from other project communities in terms of household size, farm owner gender, education levels, farm management training, and record-keeping. Maasai households averaged around nine persons while poultry households averaged around five persons. Ninety-three percent of respondents were men compared to $\approx$65% in poultry systems. Over 60% of Maasai reported having no formal education while this percentage was 0% in Zambia and Kenya, 1% in Zimbabwe and 10% in Ghana. Only 4% of Maasai households reported having training by animal health professionals or organizations (e.g. NGOs) on any aspect of farm management (record-keeping, animal health, etc.) and 10% reported keeping written records. The Maasai owned a median of 68 cattle ($Q_1$ = 20, $Q_3$ = 160) and 120 sheep and goats ($Q_1$ = 60, $Q_3$ = 255). Maasai keep these animals divided into those at the "temporary bomas" (usually distant locations to ease local

grazing pressures and cope with drought), those that move in and out of the household for daily grazing, and those kept inside the household (mostly young or sick animals). A median of 30 cattle ($Q_1 = 9$, $Q_3 = 60$) and 0 sheep and goats ($Q_1 = 0$, $Q_3 = 7$) were kept in temporary bomas. A median of 30 cattle ($Q_1 = 11$, $Q_3 = 70$) and 79 goats ($Q_1 = 35$, $Q_3 = 150$) moved in and out for daily grazing. A median of 5 cattle ($Q_1 = 1$, $Q_3 = 7$) and 6 goats ($Q_1 = 1$, $Q_3 = 8$) were kept at home. The most common self-reported diseases for cattle were Contagious Bovine Pleuropneumonia (70%), Coenurosis (63%) and East Coast Fever (61%). For sheep and goats, the top three diseases were Coenurosis (96%), Contagious Caprine Pleuropneumonia (92%) and Sheep and Goat Pox (54%).

Country specific descriptive statistics on demographics, socioeconomics, farm management practices (including antimicrobial use), and diseases reported can be found in Tables 5-Table 9 in S1 Text.

## Antimicrobial knowledge and use patterns

Across countries, 104 antimicrobial products were reported as commonly used, containing active substances representing eight antimicrobial classes. Seventy percent of these products (N = 73) contained a tetracycline and 20% (N = 45) contained a macrolide or aminoglycoside. The most common reason for using antimicrobials was for treatment followed by preventing sickness in groups and individual animals and, mostly in the Maasai, for faster and bigger growth (Fig 3). The average number of products reported by households as commonly used was around six. Farmers in Zambia and Tanzania reported the highest use of antimicrobials (≈3–5 times per month) followed by those in Zimbabwe and Ghana (1–2 times per month) and Kenya (<1–2 times per month) (see Table 8 in S1 Text). Most households acquired antibiotics from agrovet shops (≈83%) with much lower percentages (≈12% for each source) acquiring drugs from feed distributors, shops that were not agrovets (e.g., hardware stores) and government veterinarians (see Table 10 in S1 Text for pooled results and Table 11 for country specific statistics in S1 Text). When purchasing antimicrobials at agrovet shops, ≈38% reported providing symptoms of sick animals and getting advice on the specific AM (≈38%), while ≈30% were told by the agrovet which the drugs they needed but given no instructions on use and 25% told the agrovet the drugs they needed and did not receive any instructions on use (see Table 12 in S1 Text). Very few households (12%) reported almost always having a prescription when purchasing antimicrobials (see Table 13 in S1 Text for country-specific results). Observation of withdrawal was also variable with average of 35% of households consuming products at home and 20% selling the product. Around 50% of farmers reported to have observed an increase in treatment failures since they began farming (see Table 14 in S1 Text for country-specific results).

Sources of health advice related to antimicrobial use are provided in Fig 4 and individuals who administer antimicrobials are provided in Fig 5. A majority of households reported almost never or rarely asking advice from feed distributors (75%, only asked in poultry systems), private veterinarians (74%), community health workers (65%), government veterinarians (55%) and agrovets (51%) (see Table 15 in S1 Text for country-specific results). When administering antimicrobials, most respondents indicated that the farm owner almost always administered the drugs (53%), followed by the farm manager (24%) and family and friends (8%) (see Table 16 in S1 Text for country-specific results). Across countries, few households reported that government veterinarians, private veterinarians, or agrovets administered these drugs to their animals. Challenges reportedly faced when accessing these professionals included high cost of services (20%), distance to services (22%), lack of awareness of how to access these services (23%), and no or slow response (6%).

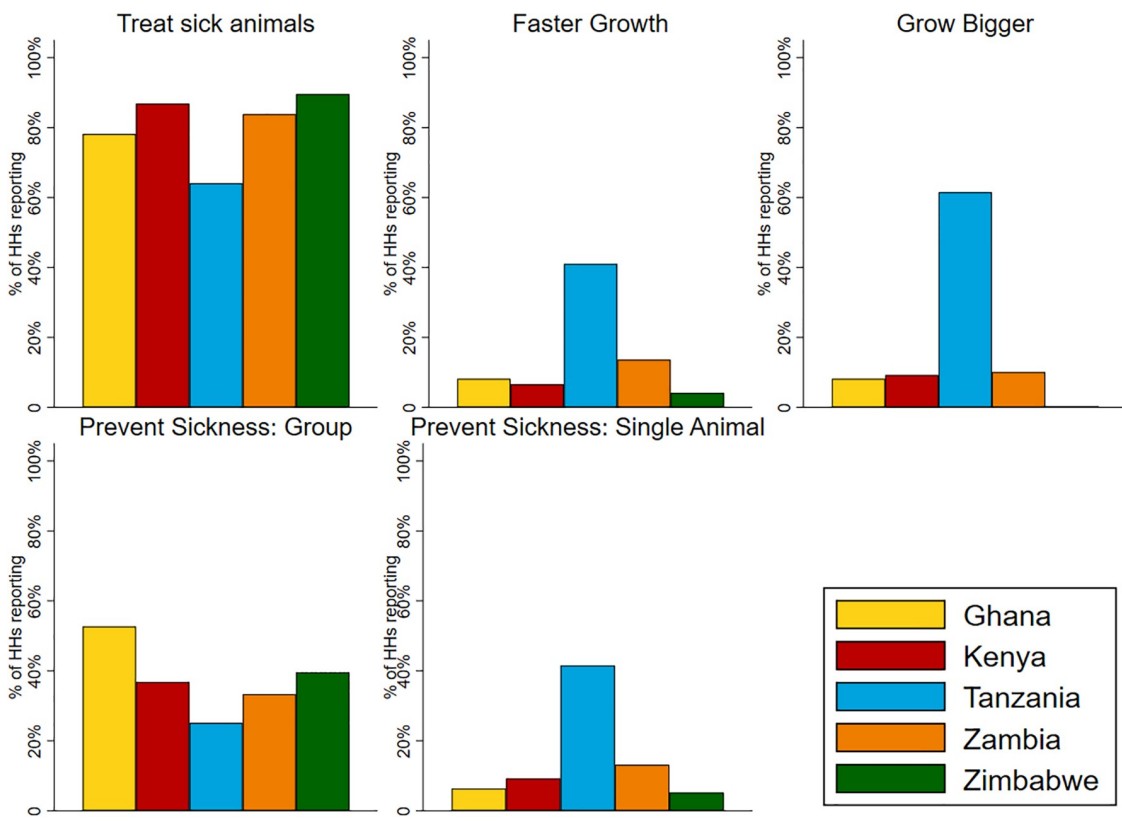

**Fig 3. The reasons farmers reported using antimicrobials in livestock across project countries.**

## Distribution of knowledge, attitudes, and practices

Fig 6 provides the distribution of the percentage of correct responses for antimicrobial use and AMR knowledge, attitudes, and practices across the project countries (see Table 17 in S1 Text for country-specific results). One Way ANOVAs showed significant differences across countries between percentages correct on knowledge ($F(4,862) = 16.89$, $p<0.0001$), attitudes ($F(4,862) = 66.26$, $p<0.0001$), and practices ($F(4,862) = 159.71$, $p<0.0001$). Pairwise comparison of mean values for knowledge and practices through Tukey-Kramer comparisons demonstrated that significant differences ($p<0.05$) across countries were largely driven by differences between pastoralist households (Tanzania) and poultry production households. While the mean correct knowledge and attitudes were around 70% in poultry system countries, the mean for Tanzania were around 40%. In contrast to knowledge and attitudes, there were significant differences in mean scores on prudent use practices across all countries, except between the broiler keeping households in Zambia and Zimbabwe, which reported the greatest adherence to prudent practices ($\approx 85\%$). See for Fig 6 –Fig 8 in S1 Text for One Way ANOVA results and Tukey and Hamer minimum significant differences for each pair of means.

## Associations between KAP across countries

Pearson's correlation was used to assess the bivariate relationship between KAP scores. Pooled across countries, KAP scales were all significantly ($p<0.05$) and positively correlated with the strongest correlations between knowledge and attitudes ($\approx 0.36$) and attitudes and practices ($\approx 0.30$) (see Table 4). Knowledge and attitudes were also positively and significantly related

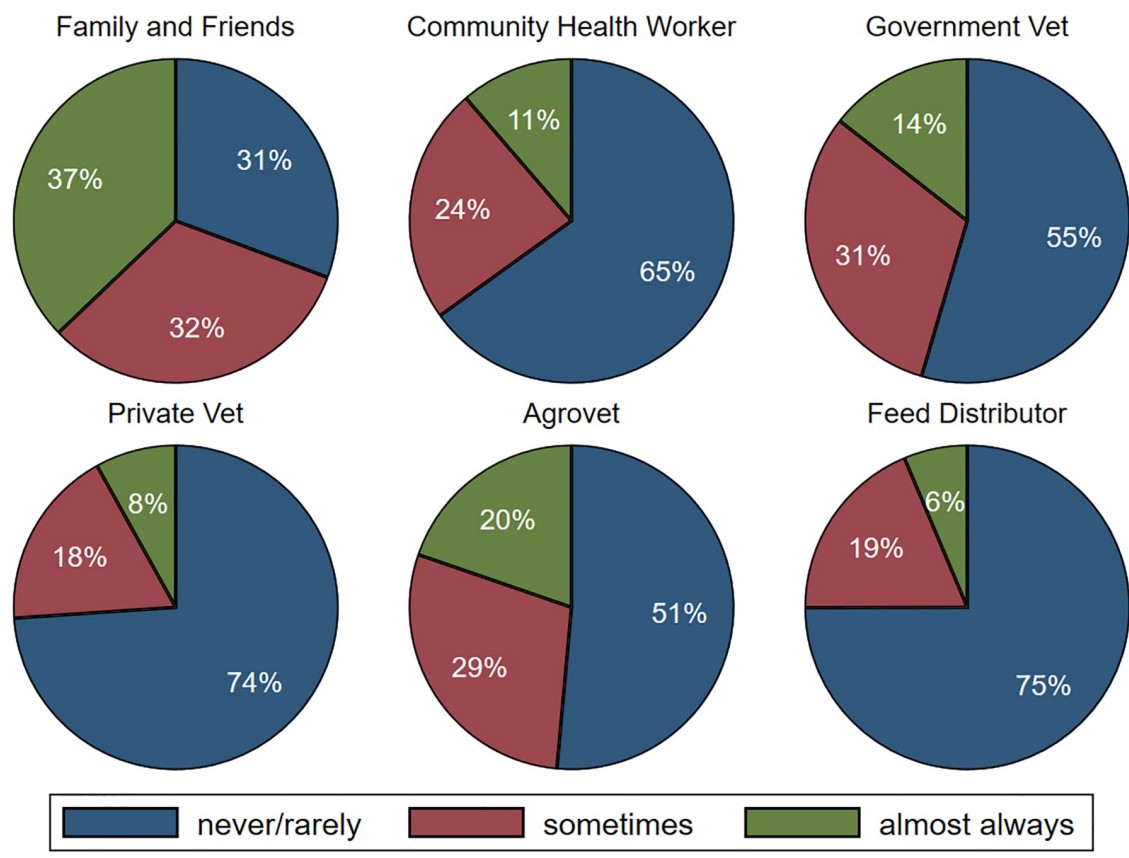

**Fig 4. Sources of advice on animal health.** N = 867 except for advice from feed distributor, which was not asked in Tanzania given the Maasai do not purchase feed for their livestock and is based on 672 observations.

within countries, ranging from ≈0.21 in Zimbabwe to ≈0.38 in Zambia. In contrast, knowledge was only significantly related to practices in Zambia (≈0.20). Likewise, attitudes were only significantly related to practices in Zambia (≈0.21) and Tanzania, although the latter in the opposite direction (≈-0.15).

## Regression analysis: KAP and demographics in poultry and pastoralist systems

Few demographic variables were significantly associated with KAP measures in poultry (Table 5, left panel) or pastoralist systems (Table 5, right panel). For poultry systems, female respondents averaged about 5% lower KAP knowledge holding other variables at their means and controlling for country. Years keeping poultry were positively associated with KAP knowledge with every additional year associated with a 0.4% increase in knowledge. The only other demographic variable associated with KAP measures was age, which was positively associated with prudent attitudes with every year increase associated with a 0.2% increase in prudent attitudes. In pastoralist systems, older individuals had less knowledge of antimicrobial use and AMR and less prudent attitudes with every year decreasing knowledge by 0.6% and attitudes by 0.2%. Finally, Maasai with tertiary education were associated with 24% higher knowledge scores.

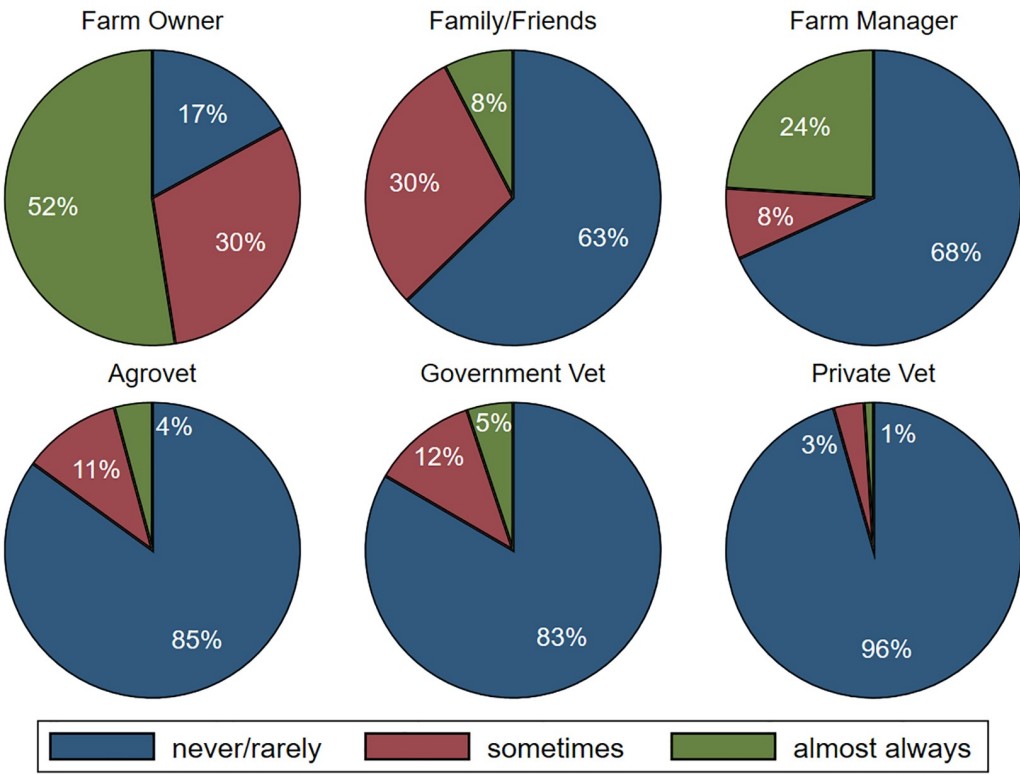

**Fig 5. People administering antimicrobials to livestock.** N = 867 except for farm manager which is based on 672 observations (Maasai generally do not have farm managers).

## Regression analysis: KAP and on-farm dynamics in poultry and pastoralist systems

In poultry systems (Table 6, left panel), antimicrobial use and AMR knowledge was higher in individuals who used more products with antimicrobials (0.2%), kept records (10.6%), and had farm training (18.4%). Poultry farmers who reported having training also had more prudent attitudes (4.2%). Those who used more products with antimicrobials had less prudent attitudes (-2.0%), although this was only marginally significant in the full model (p = 0.08, see Table 3 in S1 Text). For prudent practices, farmers who had reported increasing levels of treatment failure were associated with a 6.0% decrease in prudent practice scores although this relationship was only marginally significant in the full model (p = 0.89, see Table 3 in S1 Text). Those who received farm trainings were associated with 3.0% decrease in prudent practice scores. Poultry farmers who kept records reported 1.7% higher prudent practice scores.

In pastoralist systems (Table 6, right panel), no on-farm dynamics were related to antimicrobial use and AMR knowledge. Prudent attitudes were significantly associated with reporting increasing treatment failure, with pastoralist reporting increasing failure having 36% more prudent scores. Every standard deviation increase in farm size was associated with 2.1% greater prudent practices, while prudent practices were negatively associated with the number of diseases reported (-6.9%), increasing level of treatment failures (-44.1%) and increasing levels of antimicrobial use (-4.9%).

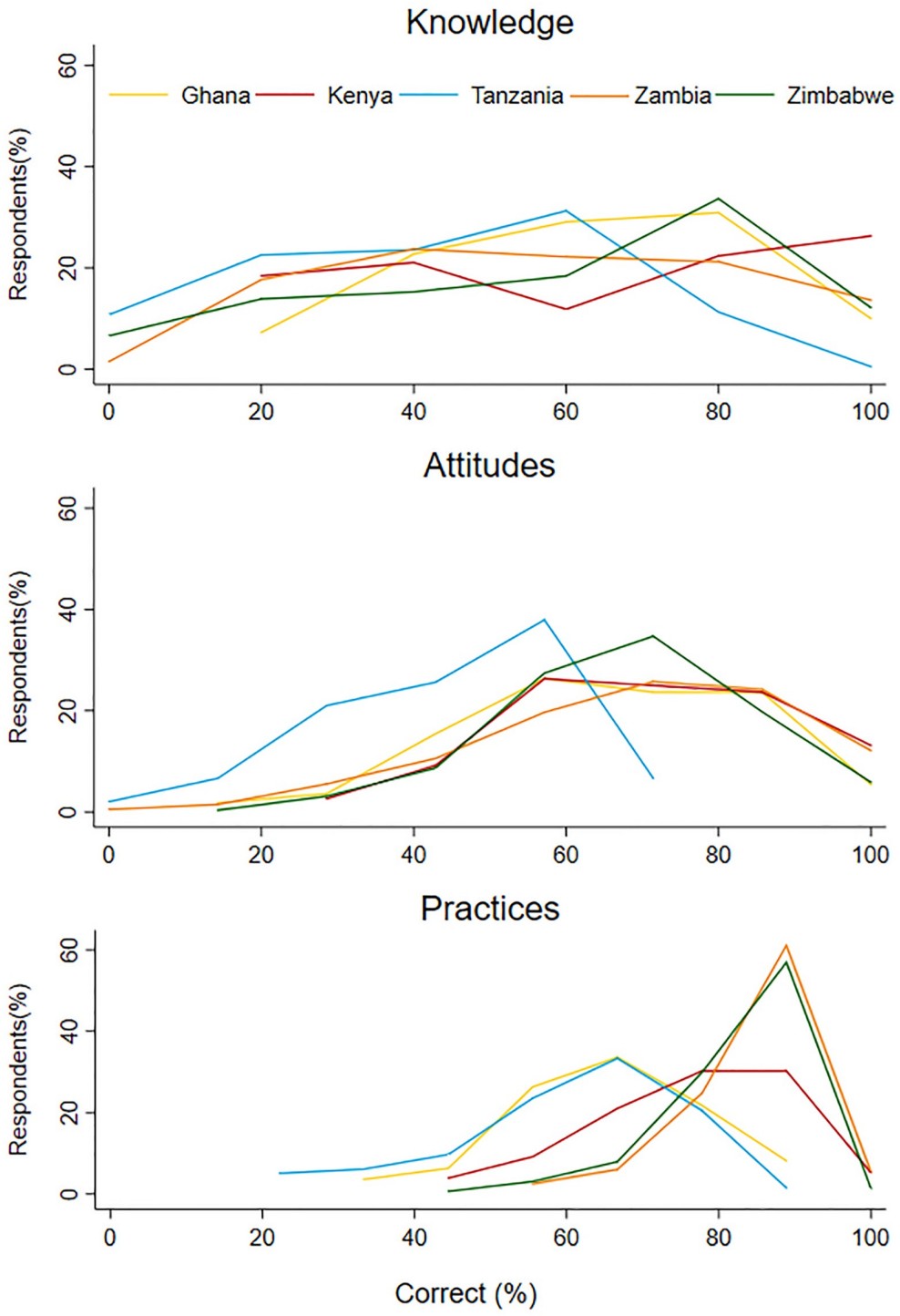

**Fig 6. The distribution of scores on antimicrobial use and AMR knowledge, attitudes, and practices scales across project countries.** The y-axis is the percentage of respondents having a certain score. The score (percentage correct) is represented on the x-axis.

**Table 4. Pearson's correlation between KAP measures across and within countries.**

| Sample | | Knowledge | Attitudes | N |
|---|---|---|---|---|
| Pooled | Attitude | 0.36*** | 1.00 | 867 |
| | Practices | 0.13*** | 0.30*** | |
| Ghana | Attitude | 0.29** | 1.00 | 110 |
| | Practices | 0.03 | 0.04 | |
| Kenya | Attitude | 0.34** | 1.00 | 76 |
| | Practices | -0.19 | 0.08 | |
| Tanzania | Attitude | 0.22** | 1.00 | 195 |
| | Practices | -0.10 | -0.15* | |
| Zambia | Attitude | 0.38*** | 1.00 | 198 |
| | Practices | 0.21** | 0.21** | |
| Zimbabwe | Attitude | 0.21*** | 1.00 | 288 |
| | Practices | 0.10 | 0.11 | |

*** p<0.001,

** p<0.01,

* p<0.05,

+ p<0.10

**Table 5. Associations between KAP measures and demographics in poultry (left) and pastoralist systems (right) adjusted for country effects.** See variable definitions in Table 3.

| VARIABLES | Poultry | | | Pastoralist | | |
|---|---|---|---|---|---|---|
| | Knowledge | Attitudes | Practices | Knowledge | Attitudes | Practices |
| Gender (1 = Female) | -0.052** | -0.001 | 0.009 | 0.071 | 0.038 | -0.040 |
| | (-0.098–-0.007) | (-0.032–0.030) | (-0.008–0.027) | (-0.058–0.200) | (-0.055–0.130) | (-0.127–0.048) |
| Age | 0.001+ | 0.002** | -0.000 | -0.006*** | -0.002* | 0.001+ |
| | (-0.001–0.003) | (0.000–0.003) | (-0.001–0.000) | (-0.008–-0.003) | (-0.003–0.000) | (-0.000–0.003) |
| Primary Education | -0.110+ | -0.003 | -0.012 | 0.050+ | -0.016 | -0.020 |
| | (-0.248–0.029) | (-0.097–0.092) | (-0.065–0.041) | (-0.022–0.121) | (-0.068–0.035) | (-0.069–0.029) |
| Secondary Education | -0.047 | 0.059 | 0.007 | 0.060 | 0.052 | 0.062+ |
| | (-0.183–0.089) | (-0.034–0.152) | (-0.045–0.059) | (-0.071–0.191) | (-0.042–0.147) | (-0.028–0.151) |
| Tertiary Education | 0.003 | 0.076+ | 0.013 | 0.240** | -0.055 | 0.037 |
| | (-0.136–0.142) | (-0.019–0.171) | (-0.041–0.066) | (0.013–0.467) | (-0.218–0.108) | (-0.118–0.192) |
| Keeping time (yrs) | 0.004*** | 0.000 | 0.001 | | | |
| | (0.002–0.007) | (-0.001–0.002) | (-0.000–0.002) | | | |
| Constant | 0.619*** | 0.567*** | 0.658*** | 0.668*** | 0.518*** | 0.546*** |
| | (0.478–0.760) | (0.471–0.664) | (0.604–0.712) | (0.544–0.791) | (0.430–0.607) | (0.462–0.630) |
| Observations | 670 | 670 | 670 | 195 | 195 | 195 |
| R-squared | 0.057 | 0.036 | 0.298 | 0.148 | 0.032 | 0.032 |

ci in parentheses

*** p<0.01,

** p<0.05,

* p<0.10,

+ p<0.2

**Table 6. Associations between KAP measures and on-farm dynamics in poultry (left) and pastoralist systems (right) adjusted for country effects.** See variable definitions in Table 3.

| VARIABLES | Poultry | | | Pastoralists | | |
|---|---|---|---|---|---|---|
| | Knowledge | Attitudes | Practices | Knowledge | Attitudes | Practices |
| Farm scale (std) | 0.025+ | -0.002 | -0.001 | -0.032+ | -0.008 | 0.021* |
| | (-0.013–0.062) | (-0.030–0.025) | (-0.017–0.014) | (-0.076–0.013) | (-0.037–0.020) | (-0.003–0.044) |
| Disease level | 0.035+ | -0.016 | -0.010 | 0.043 | 0.006 | -0.069*** |
| | (-0.008–0.077) | (-0.048–0.015) | (-0.028–0.007) | (-0.037–0.123) | (-0.046–0.057) | (-0.111–-0.028) |
| Treatment failure | 0.009 | 0.056 | -0.060* | 0.052 | 0.358*** | -0.442*** |
| | (-0.147–0.166) | (-0.058–0.171) | (-0.124–0.005) | (-0.233–0.336) | (0.175–0.542) | (-0.591–-0.292) |
| AMs used per month | -0.002 | -0.020** | 0.006 | -0.002 | -0.003 | -0.049*** |
| | (-0.029–0.025) | (-0.040–-0.000) | (-0.005–0.017) | (-0.049–0.046) | (-0.033–0.028) | (-0.074–-0.024) |
| Number AM medicines | 0.002** | 0.000 | 0.000 | 0.007 | 0.005 | -0.003 |
| | (0.000–0.003) | (-0.000–0.001) | (-0.000–0.001) | (-0.011–0.024) | (-0.007–0.016) | (-0.013–0.006) |
| Keep records | 0.106*** | 0.027+ | 0.017* | 0.061 | 0.022 | 0.008 |
| | (0.056–0.155) | (-0.009–0.063) | (-0.003–0.037) | (-0.056–0.178) | (-0.054–0.097) | (-0.054–0.069) |
| Training | 0.184*** | 0.042*** | -0.030*** | -0.061 | -0.074 | -0.039 |
| | (0.143–0.226) | (0.011–0.072) | (-0.047–-0.013) | (-0.236–0.114) | (-0.186–0.039) | (-0.131–0.053) |
| Constant | 0.305*** | 0.624*** | 0.672*** | 0.414*** | 0.354*** | 0.814*** |
| | (0.186–0.423) | (0.538–0.711) | (0.623–0.721) | (0.283–0.544) | (0.270–0.438) | (0.746–0.883) |
| Observations | 667 | 667 | 667 | 194 | 194 | 194 |
| R-squared | 0.182 | 0.037 | 0.312 | 0.033 | 0.121 | 0.340 |

## Regression analysis: KAP and sources of animal health advice in poultry farmers and pastoralists

Several sources of animal health advice were related to antimicrobial use and AMR knowledge in poultry farmers, particularly for those farmers who "almost always" sought advice from these sources (Table 7, left panel). Poultry farmers who sometimes received advice from government veterinarians were associated with 10.2% higher knowledge scores while those almost always receiving advice had 14.7% higher scores. Likewise, those sometimes and almost always receiving advice from extension officers had 5.9% and 12.6% higher knowledge scores, respectively, although these associations were not found in the full model (see Table 3 in S1 Text). Those who almost always received advice from private veterinarians had 7.9% higher knowledge scores, but this was not found in the full model (see Table 3 in S1 Text). Poultry farmers who were sometimes provided advice by agrovets had 4.4% higher scores, although this did not hold for those who "almost always" received advice, while those who almost always received advice from laboratory assistants had 17.6% *lower* scores. For prudent attitudes, the only sources of advice that retained significance were extension officers (5.5% increase for almost always), and government veterinarians, where farmers who sometimes and almost always received advice having 4.2% and 6.7% higher scores, respectively. For prudent practices, seeking advice from extension officers had a contrasting effect with those who sometimes or almost always received advice having 2.6% and 5.3% lower reported prudent practice scores. Almost always receiving advice from government or private veterinarians was associated with a 6.9% and 3.3% increase in prudent practices, respectively.

For pastoralists, those who almost always received advice from government veterinarians had 19.7% higher scores on antimicrobial use and AMR knowledge (Table 7, right panel). The marginally significant and positive relationship between sometimes getting advice from

**Table 7. Associations between KAP measures and on-farm dynamics in poultry (left) and pastoralist systems (right) adjusted for country effects.** See variable definitions in Table 3.

| VARIABLES | Poultry | | | Pastoralist | | |
| --- | --- | --- | --- | --- | --- | --- |
| | Knowledge | Attitudes | Practices | Knowledge | Attitudes | Practices |
| **Agrovet_advice** | 0.044* | 0.007 | 0.003 | 0.056+ | -0.003 | 0.001 |
| - Sometimes | (-0.004–0.091) | (-0.026–0.040) | (-0.015–0.021) | (-0.030–0.142) | (-0.059–0.053) | (-0.052–0.054) |
| **Agrovet_advice** | 0.016 | 0.020 | -0.017+ | -0.003 | -0.033 | 0.049 |
| - Almost always | (-0.042–0.074) | (-0.021–0.060) | (-0.040–0.005) | (-0.134–0.128) | (-0.118–0.053) | (-0.032–0.130) |
| **Extension advice** | 0.059* | 0.034+ | -0.026** | 0.032 | 0.085*** | -0.054** |
| - Sometimes | (-0.005–0.122) | (-0.010–0.078) | (-0.050–-0.001) | (-0.047–0.111) | (0.033–0.137) | (-0.103–-0.005) |
| **Extension advice** | 0.126*** | 0.055** | -0.053*** | 0.001 | -0.050 | 0.009 |
| - Almost always | (0.055–0.197) | (0.005–0.105) | (-0.080–-0.025) | (-0.243–0.246) | (-0.210–0.110) | (-0.142–0.160) |
| **Gov_vet advice** | 0.102*** | 0.042** | 0.002 | -0.009 | 0.014 | -0.054** |
| - Sometimes | (0.046–0.158) | (0.003–0.081) | (-0.019–0.024) | (-0.088–0.071) | (-0.038–0.066) | (-0.104–-0.005) |
| **Gov_vet_advice** | 0.147*** | 0.067*** | 0.069*** | 0.197** | 0.045 | -0.090* |
| - Almost Always | (0.082–0.211) | (0.021–0.112) | (0.044–0.094) | (0.037–0.356) | (-0.059–0.150) | (-0.188–0.009) |
| **Priv_vet advice** | 0.016 | -0.048+ | 0.003 | | | |
| - Sometimes | (-0.073–0.105) | (-0.111–0.014) | (-0.032–0.037) | | | |
| **Priv_vet_advice** | 0.079** | -0.018 | 0.033** | | | |
| -Almost always | (0.003–0.156) | (-0.072–0.035) | (0.003–0.063) | | | |
| **Laboratory_advice** | 0.022 | -0.041 | 0.006 | -0.091 | 0.117** | -0.047 |
| - Sometimes | (-0.082–0.126) | (-0.114–0.032) | (-0.035–0.046) | (-0.265–0.083) | (0.003–0.231) | (-0.155–0.060) |
| **Laboratory_advice** | -0.176* | -0.033 | -0.038 | -0.222 | -0.063 | -0.263* |
| - Almost always | (-0.363–0.011) | (-0.164–0.098) | (-0.111–0.034) | (-0.707–0.262) | (-0.380–0.254) | (-0.561–0.036) |
| **Friends_advice** | 0.010 | 0.009 | -0.002 | 0.060 | -0.058 | -0.000 |
| -Sometimes | (-0.038–0.058) | (-0.025–0.043) | (-0.021–0.016) | (-0.109–0.228) | (-0.168–0.052) | (-0.104–0.104) |
| **Friends_advice** | 0.020 | -0.009 | -0.002 | 0.025 | -0.007 | -0.051 |
| -Almost always | (-0.033–0.072) | (-0.046–0.027) | (-0.022–0.018) | (-0.129–0.178) | (-0.107–0.093) | (-0.146–0.043) |
| Constant | 0.470*** | 0.603*** | 0.645*** | 0.365*** | 0.413*** | 0.701*** |
| | (0.401–0.539) | (0.555–0.652) | (0.619–0.672) | (0.216–0.515) | (0.316–0.511) | (0.609–0.793) |
| Observations | 672 | 672 | 672 | 195 | 195 | 195 |
| R-squared | 0.113 | 0.049 | 0.342 | 0.057 | 0.115 | 0.122 |

agrovets and knowledge (5.6% increase) was significant and higher in the full model (8.7%) (see Table 4 in S1 Text). For prudent attitudes, those who sometimes received advice from extension officers and laboratory personnel had 8.5% and 11.7% higher scores, respectively. In general, receiving animal health advice was negatively associated with prudent practice scores in pastoralists with those who sometimes received advice from extension officers and government veterinarians associated with 5.4% lower scores. Those who almost always received advice from government veterinarians had 9.0% lower reported prudent practice scores and those almost always receiving advice from laboratory personnel had 26.3% lower scores. However, the only source of advice to maintain significance in the full model was seeking advice from laboratory personnel (see Table 4 in S1 Text).

## Discussion

In this study we examined the knowledge, attitudes, and practices related to antimicrobial use and resistance in livestock farmers across five African countries (Ghana, Kenya, Tanzania, Zambia, Zimbabwe). Broadly, we found that it is individuals who live or work at the farm who

draw upon their knowledge and experiences to make decisions regarding animal healthcare, most often with input from family, friends and neighbors. It is also these individuals who are responsible for administering antimicrobials to their animals, mostly for therapeutic reasons in poultry systems and more often for growth promotion and disease prevention among pastoralists. Reliance on animal health professionals for advice and treatment antimicrobials is very limited. When purchasing antimicrobial products, most respondents said they relied upon local agrovet drug shops and did so largely without prescriptions. Observation of withdrawal periods after treatment with antimicrobials was also limited with farmers both consuming and selling animal products (meat, milk, eggs) from animals currently undergoing treatment or still within the withdrawal period. High levels of engagement in the 'informal veterinary sector'–characterized by 'lay' diagnoses and treatment, limited input from trained health professionals, and non-prudent practices–are consistent with findings from other studies conducted in the project countries including among Maasai pastoralists in Tanzania and Kenya [28,32,38,39], cattle keepers in Eastern Province, Zambia [40] and poultry farmers in Ghana, Kenya, Tanzania, Zambia, and Zimbabwe [28–37].

Collectively, these results further confirm the negative impacts of the Structural Adjustment Policies that relegated the public veterinary services in many LMICs to policy and regulation functions [45]. The few veterinarians who went into private practice were often located in areas where they could make good returns, which often meant they were removed from farmers in rural areas, especially those inhabiting arid environments such as pastoralists. As our results suggest, this shift towards paid veterinary services has continued to limit access to animal health professionals with gaps in veterinary care being filled by potentially unqualified persons, such as employees at agrovets. These individuals, while doubling as drug shop agents and service providers, often have limited levels of formal training in animal health and knowledge on AMR and prudent antimicrobial use. Agrovet shop owners are also under immense pressure to sell medicines to farmers at the risk of losing their livelihood if they fail to meet consumer demands.

Our results further suggest that limited access to animal health professionals, with subsequent engagement in the informal veterinary sector, may become a priority issue for National Action Plans on AMR in resource-limited countries, and that intervention strategies aimed at boosting prudent drug use will strongly benefit from contextualizing practices at the farm level. Indeed, our results highlight the value of stakeholder analysis as part of any practical solution given significant differences in sociocultural, economic, environmental and historic factors. For example, farmers in poultry systems in our study could recount significantly greater amounts of AMR knowledge and reported more prudent attitudes and practices compared to surveyed Maasai pastoralists. These differences may be due to varying educational backgrounds as poultry farmers tended to be better educated on average, although we only found a significant positive relationship between education level and KAP scales in the Maasai at the highest level of education (see Table 3). These differences could also be driven by variations in indigenous ethnoveterinary belief systems, which the Maasai draw heavily upon to diagnosis and treat their animals. If these beliefs are inconsistent with scientific concepts underlying AMR (e.g., selection, transmission), then some Maasai may not perceive AMR as risky or believe non-prudent use could adversely impact on their livelihoods [53]. Another consequence of this enduring belief system may be that, compared to poultry systems surveyed in this study, input from animal health professionals does not have a large influence on Maasai knowledge, attitudes, and practices. Indeed, these inputs were not significantly related to KAP measures in the Maasai (Table 5). More generally, establishing the relative contributions of the different livelihood factors on knowledge, attitudes and practices related to antimicrobial use

and AMR is critical in resource-constrained contexts such as LMICs where only a limited number of factors can be targeted within behavioral change interventions.

## Integrating top-down support with bottom-up behavior change initiatives as a way forward

Behavior change strategies based primarily on top-down approaches, including new regulations, are less likely to result in broad and enduring changes in antimicrobial use within LMICs given limitations in existing regulatory capacities and animal health services, as well as a tendency for such top-down approaches to fail to identify and account for stakeholder preferences and concerns. Although top-down initiatives might be delivered at scale more quickly and may be easier to control in a context of strong monitoring and enforcement, these approaches tend to require more resources, may generate impractical solutions with unintended consequences, and can be more difficult to sustain. By contrast, bottom-up stakeholder support programs benefit from being co-designed by the target stakeholders themselves to render them more practical (for stakeholders to implement) and more effective and sustainable. The bottom-up approach can be more resource efficient as well when leveraging smaller 'seed' stakeholder groups for a change program, which then diffuses knowledge and new practices through professional and social networks to bring other stakeholders on board. As long as there is a demonstrable benefit justifying the shift in practice–for instance, more reliable production output over several years with reduced losses due to improved biosecurity practices and more careful targeting and delivery of antibiotic prescriptions–other stakeholders are more likely to embrace the risk of trying something new.

Our results provide insights for the design of these potential interventions. For example, we show that agrovet employees are often the only stakeholders outside the household with an animal health background that farmers interact with between deciding on the need for treatment and the administration of antimicrobials. However, our regression models showed that those who received animal health advice from agrovets had less knowledge and less prudent attitudes and practices (see Table 5). Combined, these findings suggest that agrovets could play an important role in providing information on antimicrobial use and AMR to farmers unable to access a veterinarian but that training of agrovet employees on AMR and the importance of prudent use practices is likely a first requirement. Importantly, and as our results among farmers suggest, solely providing information on antimicrobial use and AMR to agrovets is unlikely to promote more prudent dispensing practices. Indeed, in a survey of agrovets in Africa and Asia, knowledge of AMR did not translate into prudent use (e.g., reduced dispensing) but was linked to *less* prudent practices (i.e., use of next-line antibiotics)[54]. In addition, the double role played by agrovets–those who give advice that leads to antimicrobial use and those who stand to benefit financially from the sales of antimicrobials–gives rise to moral hazard that must be considered when designing interventions. [55]. To understand how to motivate prudent dispensing practices among agrovets [39,56], more research is needed on their knowledge, attitudes, and practices regarding antimicrobial use and AMR.

Finally, while our results suggest the need for bottom-up strategies to address AMR at the individual provider or user level, there remains a clear need for broader infrastructural changes in resource-limited countries (e.g., strengthening the veterinary workforce and better resourced food safety departments) to remove hurdles and provide an enabling environment for individual-level change [57]. Our models of health-seeking practices, for example, demonstrated the positive role that the professional veterinary sector can play in promoting prudent drug use practices. Indeed, poultry farmers, who more frequently sought advice from professional veterinarians, reported more prudent knowledge and attitudes and better practices

overall (Table 5). Considering these trends, intervention strategies targeted at veterinarians (e.g., stewardship programs) *could* prove successful if they were coupled with broader infrastructural changes to increase access to animal health professionals. Importantly, while top-down strategies are needed, they must still be informed by the ground-level realities of antimicrobial use and AMR challenges, or they may risk doing more harm than good. Wholesale application of some popular top-down regulations in LMICs (e.g., restrictions on certain types of antimicrobials) risks preventing access to antimicrobials in contexts where a lack of access to these drugs continues to result in the deaths of considerably more people and animals than resistant infections [4,24]. Therefore, we recommend participatory and stakeholder-led processes for the development of more practical, robust and sustainable solutions to combat AMR.

## Study limitations and future directions

This study represents one of the most comprehensive surveys of knowledge, attitudes, and practices related to antimicrobial use and AMR across livestock systems in Africa. However, more robust ethnographic work is needed across cultures and production systems and scales to identify the factors associated with KAP variation. While we found that demographic, livelihood and health-seeking factors were significantly related to KAP measures, these factors usually accounted for low levels of variance (<5%). Most of the variance was accounted for by country indicators, especially for practices, which significantly varied across countries and clearly hold the greatest consequences for development and transmission of AMR. The contribution of country-level indicators suggests the existence of factors (cultural, historical, economic) that were not, or could not be, recovered in our mostly self-report cross-sectional surveys. Identification of some of these factors will require different study designs (i.e., longitudinal) and ethnographic techniques (participant-observation). Observational methods are warranted given evidence of self-desirability bias in the reporting of prudent practices. Indeed, while our focus group participants often confessed to not following withdrawal periods, our KAP results indicated that up to 30% of households reported observing withdrawal periods. Recall biases also impact the collection of antimicrobial use data and we only used self-report measures to quantify antimicrobial use. Methods such as repeated self-report measures, passive surveillance methods, including collection of used sachets and bottles in waste buckets, and triangulation with sales data at agrovets will be needed to produce accurate and quantitative antimicrobial use data.

Critically, intervention strategies must also reconcile the lack of association between practices and knowledge and attitudes in most countries. Especially in poultry systems, the observance of non-prudent practices cannot be explained by a lack of knowledge or appropriate attitudes concerning AMR. Around 70% of respondents held sufficient knowledge to understand the AMR issue and associated desirable attitudes but this was not reflected in their practices. Longitudinal research will also be needed to establish the direction of causality between related knowledge, attitudes, and practices. For example, are those who know about the impacts of AMR less likely to experience treatment failure (e.g., through visiting veterinarians, getting diagnostics) or do those who experience treatment failure learn about AMR through the process of dealing with the failure? Understanding the directionality of these relationships will be critical for crafting intervention strategies in the future.

## Conclusion

This study has shown that livestock farmers in five African countries make decisions on antimicrobial use with limited (or no) inputs from animal health professionals, who are often

inaccessible due to distance and/or financial considerations. Given these realities, interventions to promote prudent use practices in the short-term must focus on the contexts where animal healthcare decisions are made–at the farm level–and must be delivered using channels that have been demonstrated to be impactful if they are to make a difference. Until the livestock sectors within these countries support sufficient numbers of veterinarians providing quality and accessible services, including provision of accurate information on prudent drug use practices, fellow community members and agrovet shop workers will continue to lead in providing veterinary advice and care. Therefore, any interventions aimed at optimizing use of antimicrobials must consider the roles of the agrovets and community members in helping to shift practices in farming. These interventions should be founded upon a bottom-up approach that identifies the knowledge, attitudes, and practices patterning drug use. Conducting these studies across cultural and production contexts will provide the much-needed evidence base to develop and implement targeted behavioral change interventions to reduce AMR globally.

## Supporting information

**S1 Text. Study location, sampling strategies and supplemental tables, ANOVA, and regression analysis.**
(DOCX)

**S1 File. KAP survey.**
(PDF)

**S2 File. Data.**
(XLS)

## Author Contributions

**Conceptualization:** Mark A. Caudell, Alejandro Dorado-Garcia, Suzanne Eckford, Chris Creese, Denis K. Byarugaba, Kofi Afakye, Folorunso O. Fasina, Emmanuel Kabali, Stella Kiambi, Tabitha Kimani, Geoffrey Mainda, Peter E. Mangesho, Kululeko Dube, Bachana Rubegwa.

**Data curation:** Mark A. Caudell, Stella Kiambi.

**Formal analysis:** Mark A. Caudell, Denis K. Byarugaba, Kofi Afakye, Tamara Chansa-Kabali, Emmanuel Kabali, Stella Kiambi, Peter E. Mangesho, Kululeko Dube, Tendai Mugara.

**Funding acquisition:** Suzanne Eckford.

**Investigation:** Mark A. Caudell, Denis K. Byarugaba, Kofi Afakye, Stella Kiambi, Geoffrey Mainda, Peter E. Mangesho, Francis Chimpangu, Kululeko Dube, Eric Koka, Tendai Mugara, Bachana Rubegwa, Samuel Swiswa.

**Methodology:** Mark A. Caudell, Alejandro Dorado-Garcia, Suzanne Eckford, Chris Creese, Kofi Afakye, Tamara Chansa-Kabali, Folorunso O. Fasina, Emmanuel Kabali, Stella Kiambi, Geoffrey Mainda, Peter E. Mangesho, Francis Chimpangu, Kululeko Dube, Bashiru Boi Kikimoto, Eric Koka, Tendai Mugara, Samuel Swiswa.

**Project administration:** Mark A. Caudell, Alejandro Dorado-Garcia, Suzanne Eckford, Chris Creese, Kofi Afakye, Emmanuel Kabali, Tabitha Kimani.

**Visualization:** Mark A. Caudell.

**Writing – original draft:** Mark A. Caudell, Alejandro Dorado-Garcia, Suzanne Eckford, Chris Creese.

**Writing – review & editing:** Mark A. Caudell, Alejandro Dorado-Garcia, Suzanne Eckford, Chris Creese, Denis K. Byarugaba, Kofi Afakye, Tamara Chansa-Kabali, Folorunso O. Fasina, Emmanuel Kabali, Stella Kiambi, Tabitha Kimani, Geoffrey Mainda, Peter E. Mangesho, Francis Chimpangu, Kululeko Dube, Bashiru Boi Kikimoto, Eric Koka, Tendai Mugara, Samuel Swiswa.

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
