## [Decision Letter · Decision Letter 0]

11 Sep 2019

PONE-D-19-19328

Towards a bottom-up understanding of antimicrobial use and resistance on the farm: A knowledge, attitudes, and practices survey across livestock systems in five African countries

PLOS ONE

Dear Dr. Caudell,

Thank you for submitting your manuscript to PLOS ONE. After careful consideration, we feel that it has merit but does not fully meet PLOS ONE’s publication criteria as it currently stands. Therefore, we invite you to submit a revised version of the manuscript that addresses the points raised during the review process.

The study is very well presented, but lacks in the quality of preparation. I agree with reviewers. The main problem found in the manuscript is related to the some aspects of methodology, confusing results, tables and redaction style. Please review the referee comments and make your peer revision.

We would appreciate receiving your revised manuscript by Oct 26 2019 11:59PM. To enhance the reproducibility of your results, we recommend that if applicable you deposit your laboratory protocols in protocols.io, where a protocol can be assigned its own identifier (DOI) such that it can be cited independently in the future. For instructions see: http://journals.plos.org/plosone/s/submission-guidelines#loc-laboratory-protocols

We look forward to receiving your revised manuscript.

Kind regards,

Arda Yildirim, Ph.D.

Academic Editor

PLOS ONE

Journal Requirements:

2. We note that you have reported significance probabilities of 0 or <0 in places. Since p=0 is not strictly possible, please correct this to a more appropriate limit, eg 'p<0.0001'.

3. In your Methods section, please provide additional location information, including geographic coordinates for the data set if available.

4. Please provide additional details regarding participant consent. In the ethics statement in the Methods and online submission information, please ensure that you have specified whether consent was suitably informed.

5. We note that Figures 2 and Supporting Information Figures]in your submission contain map/satellite images which may be copyrighted. All PLOS content is published under the Creative Commons Attribution License (CC BY 4.0), which means that the manuscript, images, and Supporting Information files will be freely available online, and any third party is permitted to access, download, copy, distribute, and use these materials in any way, even commercially, with proper attribution. For these reasons, we cannot publish previously copyrighted maps or satellite images created using proprietary data, such as Google software (Google Maps, Street View, and Earth). For more information, see our copyright guidelines: http://journals.plos.org/plosone/s/licenses-and-copyright.

You may seek permission from the original copyright holder of Figures [2 and Supporting Information Figures] to publish the content specifically under the CC BY 4.0 license. 

If you are unable to obtain permission from the original copyright holder to publish these figures under the CC BY 4.0 license or if the copyright holder’s requirements are incompatible with the CC BY 4.0 license, please either i) remove the figure or ii) supply a replacement figure that complies with the CC BY 4.0 license. Please check copyright information on all replacement figures and update the figure caption with source information. If applicable, please specify in the figure caption text when a figure is similar but not identical to the original image and is therefore for illustrative purposes only.

Additional Editor Comments (if provided):

For your guidance, you can check the reviewers' comments.

Reviewers' comments:

Reviewer's Responses to Questions

**Comments to the Author**

1. Is the manuscript technically sound, and do the data support the conclusions?

Reviewer #1: Partly

Reviewer #2: Partly

Reviewer #3: Partly

Reviewer #4: Yes

Reviewer #5: Partly

Reviewer #6: Yes

Reviewer #7: Partly

Reviewer #8: Yes

2. Has the statistical analysis been performed appropriately and rigorously? 

Reviewer #1: Yes

Reviewer #2: Yes

Reviewer #3: Yes

Reviewer #4: I Don't Know

Reviewer #5: No

Reviewer #6: Yes

Reviewer #7: No

Reviewer #8: Yes

3. Have the authors made all data underlying the findings in their manuscript fully available?

Reviewer #1: Yes

Reviewer #2: Yes

Reviewer #3: Yes

Reviewer #4: No

Reviewer #5: No

Reviewer #6: Yes

Reviewer #7: Yes

Reviewer #8: Yes

4. Is the manuscript presented in an intelligible fashion and written in standard English?

Reviewer #1: Yes

Reviewer #2: Yes

Reviewer #3: Yes

Reviewer #4: Yes

Reviewer #5: No

Reviewer #6: Yes

Reviewer #7: Yes

Reviewer #8: Yes

5. Review Comments to the Author

Reviewer #1: The authors made a good attempt to evaluate the knowledge, attitudes, and practices related to antimicrobial use and resistance in livestock farmers. The paper is interesting and well written. However, there are some suggestions for its improvement. In Introduction, a separate literature review section may be added. The discussion lacks strong support from previous studies which should be checked.

Reviewer #2: Abstract: The abbreviation AMR is used in the abstract but is not defined.

General comments:

There are a lot of abbreviations used in this paper. In my opinion this is very distracting. I am fairly sure the full version of AMU is never given. Please limit the use of abbreviations, and make sure all remaining abbreviations are defined at least once.

Ethnographic approach was including in the abstract but not addressed in the survey development. It is again discussed as a limitation. The definition of this approach and how the distribution of this survey either meets, or does not meet this approach is needed.

More information regarding the types of questions included in the survey instrument are needed. Questions are discussed in the analysis, but the general outline of the survey instrument is not provided in the materials and methods.

More information regarding the methodologies used are needed. The name of tests and methods are often given without citations, explanations for use, or interpretations are given.

It seems that the term antimicrobial and antibiotic are used somewhat interchangeably in this paper. I do not think these two terms are equivalent, definitions are needed to point out the differences if you are using them intentionally.

It seems odd to describe the variables used in models as bullet points below a table. The models should be better defined in the materials and method section. I could not follow the tables. I wasn’t sure which results were from the OLS models, etc. The models should be clearly written out with all variables explained in the materials and methods section. Are you sure OLS is the right approach? It looks like to me you ran three models: knowledge, attitude, practices. Is it possible that these models have correlated error terms? In this case a seemingly unrelated regression would be more appropriate. This could be incorrect, but as the models are not defined right now it is unclear to me that OLS is the correct model.

Materials and Methods: What is Kobo Collect? A software program that hosted the survey instrument? Please define.

Please define and explain how the survey approach met the requirements of a purposive sampling approach.

Tukey-Kramer comparison- This method needs further explanation and a citation. How are the results interpreted?

Table 1: From my understanding the recoded values is used as an indicator of the “right” answer. For the statement “Giving animals antimicrobials can help them GROW BIGGER…” answering disagree was coded with a 1. If the animal was sick giving antimicrobials would potential result in these positive outcomes. Additionally, there is evidence of increased production with the use of sub therapeutic doses of antimicrobials, I understand this is not a “good” practice, but if we are simply considering correct vs incorrect the statement is actually correct.

I have a similar issue with the statement “Using vaccines can reduce use of antibiotics.” Vaccines can be for viral (more common) or antimicrobial issues. So depending on which vaccine the respondent was thinking of, or more familiar with the “right” answer changes.

Cook’s D- this method needs a citation as well as an explanation.

“..these controls were not collinear with variables…” how was this determined?

Shapiro-Wilk Test- ciation is needed? What is the interpretation of this test?

Antimicrobial knowledge and Use Patterns:

How is a treatment failure defined?

KAP and on-farm dynamics

Table 4 is mentioned in the text but is missing. I think table 5 should be re-labeled.

What is meant by correlates in Table 6? What method was used to obtain these results?

Editorial:

Check for missing commas throughout to improve understanding.

Pg 5 line 117 missing period after [8,9]

Reviewer #3: I enjoyed reading this paper. It is a well written descriptive analysis of the knowledge, attitudes and practices regarding antimicrobial use and antimicrobial resistance in livestock systems among households keeping layers in Ghana and Kenya, pastoralists keeping cattle, sheep and goats in Tanzania, and broiler farmers in Zambia and Zimbabwe. The analysis seems competently executed although I have some concerns regarding comparability across countries, endogeneity, and the interpretation of some results (details below).

The authors do a good job motivating the need to understand the factors behind the misuse and possible abuse of antimicrobial drugs in LMI countries, but it was not clear why they decided to focus on these specific 5 countries (Ghana, Kenya, Tanzania, Zambia and Zimbabwe) and livestock systems. I would like to see a detailed explanation of why they chose those specific countries/systems in Africa, and why not also other LMICs in Asia or Latin America.

Within each country-livestock system, the choice of regions in the sampling frame was well justified (namely, regions where that livestock system was most prevalent in the country). However, the sampling strategy felt disjointed, as if the 5 projects were independent and glued together in the last minute. For example, in the description of study locations and sampling in the supplement, much more detail (and additional information) is offered for Tanzania and Zimbabwe than for other countries. More fundamentally, I am concerned about the comparability and “poolability” of results. While the regression analysis addresses some of this concern by including country dummies, I wonder if there are also country-specific associations with some of the independent variables included in the regression (e.g. the coefficient on the variable farm scale, or education, or agrovet advice could be country specific). To alleviate this concern this, in addition to splitting the sample between pastolarists and everybody else, I would like to see the results of the regressions for each individual country. Perhaps they are very similar and can be pooled, but it would be good to check if that is the case. The same apply for the descriptive statistics. While for the sake of conciseness it may make sense to report descriptive stats for the whole sample, if there were marked differences across countries that wouldn’t be advisable. Those differences should be highlighted.

The key dependent variables, knowledge, attitudes and practices are each a weighted average of a number of underlying variables. While I can see the value in doing this, I also wonder about explaining some of the individual components that may be particularly important for the fight against antimicrobial misuse – for example, if giving animals antimicrobials to make them fatter in Tanzania is identified as a key misconception, what are the factors explaining this?

There are many typos in the presentation of the descriptive results; please revise the numbers carefully. Some of the ones I’ve caught:

- Line 288: should be 48 (not 41) percent.

- Line 289: 22 (instead of 21) percent.

- Line 290: Zimbabwe instead of Zambia.

- Line 301: Maasai households average 9 persons

- Line 302: Ninety-three percent “of respondents” (not “of farm owners”) – since not all respondents are farm owners.

- Line 304: Swap Kenya and Zimbabwe.

- In Table S2 and S3 there are some inconsistencies. The frequencies in Table S3 do not add up to those in Table S2.

- Line 343: 65 (not 63) per cent

…

The education variable is treated as a linear variable. However, moving from a 0 (none) to 1 (primary), is not the same than moving from a 1 to a 2 (secondary). I would recommend including education as a set of dummies for primary, secondary, and tertiary and above, where “none” is the reference category.

The authors estimate three different regression models to explain knowledge, attitudes and practices regarding antimicrobial use and resistance; each with a different set of regressors: demographics, farm characteristics and antimicrobial use, and sources of animal health advice. Why not estimate a single, comprehensive model including all the explanatory variables? This would help alleviate the risk of omitted variable bias, especially if some of the regressors are correlated.

In the regression models, some of the independent variables are arguably caused by the variables they are meant to explain. I am thinking specifically about treatment failure (which is also one of the most important implications of microbial misuse). The finding that around 50% reported to observe an increase in treatment failures is interesting and important by itself, and worth being further explored – perhaps in a regression analysis.

Regarding the policy implications of some of your results: In some countries, antimicrobials are cheaper when purchased with a prescription. Is this the case in the countries you are analyzing? The fact that only 12% of the households use prescriptions regularly suggest that’s not the case, or that the price differential does not justify seeking a prescription. Could you please elaborate on the merits of such policy (and others that are being used in high-income countries) in your context?

From the questionnaire, it is not clear if the limited access to animal health professional is by choice or it is a constraint. Questions 83 and 85 only ask about where respondents get information/advice, and how often certain groups people give them advice, but are mute as to whether they face challenges getting advice from their favorite source. Question 100 captures these challenges more directly and I would recommend that the authors report the statistics of this variable more prominently.

I do not think that results reported in the paper “stress that intervention philosophies based primarily upon top-down approaches are unlikely to result in broad changes in AMU within LMICs” lines 531-533. The paper simply doesn’t ask this question. (Moreover, at other points in the paper the authors point at the negative impact of the structural adjustment policies that reduced public veterinary services on AMU– They cannot have it both ways).

Regarding interventions aimed at agrovets (line 541 on), another factor to consider is the possible lack of “incentive compatibility”, that is, as agrovets profit from the sale of AMs, they may have an incentive to recommend overuse.

I do not think that your results emphasize the importance of bottom-up strategies to address AMU and AMR (lines 569-570) – The study documents dimensions of knowledge, attitudes and practices related to AMU and AMR in livestock systems, and identifies the livelihood factors associated with these dimensions (lines 54-56). Sure, all of this is important to inform bottom up interventions, but claiming that your results support those bottom up strategies is going too far.

Other comments:

- Line 109 and 111: Please define One Health more explicitly.

- Lines 191-192: would this include veterinarians as a separate category from district veterinary officers and community animal health workers?

- Line 263: should be “withot” not “with”

- Line 298: Spell out CRD. I would also recommend give a brief explanation of what these diseases entail.

- Table S4: The variable Read does not belong with the type of livestock, I would move it to the section that includes education level variables.

- Line 327-328: present these variables in tables of descriptive statistics.

- Lines 356: I do not know how to interpret this figure. Is it supposed to be a pdf or a histogram? Can you please explain?

Reviewer #4: • In the AMR abstract please specify …………… (line 72)

• At the end of the introduction, please state the objective of this study

• In the method of writing a more specific regression model for example Y = f (X1, X2, ...) so that it can explain the

model in Tables 3, 4, 6.

• In the method not mentioned Pearson's Correlation analysis tool, the results suddenly appear in Table 2

• In the "results" Table 5 ...?

Reviewer #5: I would like to thank the authors for the good paper.

Comments

- The outputs of the focus groups discussion and Key informant Interviews should be presented in the paper

- Literature review on the topic needs to be presented

- The developed conceptual model is not based on theories or empirical work

- what is the difference between KAP measures and constructs? did you carry out factorial analysis where is the output of the analysis?

- the output of the OLS analysis is not presented> why re-coding of the scales is necessary since it has decreased information?

- the paper requires a lot of improvements in terms of methodology and analysis

Reviewer #6: Excellent manuscript on an important, timely, and interesting topic.

(1) The major results of the research is the level of knowledge, attitude, and practice (KAP) across producers in Africa. Could the manuscript be improved by including tables of sample means for the variables defined in table 1 for each country? Most of the important results, discussion, and implications have to do with these summary statistics... the regressions do add information, but the sample means are important to understanding what is going on.

(2) The comparison across practices (layers, broilers, and cattle) may confound the regression results. The reasons for using antimicrobials are much different across practices (Figure 3), the dosages are much different... a table of means would help here, as would adding a variable for practice type in the regressions... partly captured by country variables, but may want to add practice variables.

(3) Is there a conflict of interest between the antimicrobial supplier and the producers? Is it in the best interest of the suppliers to maximize sales? How does this affect KAP scores? How does that affect education efforts?

(4)These results indicate that producers in five African nations might use "too much" input... in other research, African producers may not use "enough" medicine... (eg. Marsh, Foot and Mouth Disease). Is there an explanation between these types of results?

Great work, fascinating manuscript, very well done.

Reviewer #7: This is an interesting and important topic The manuscript is well written and for the most part easy for the reader to follow. There are two very important problems to be fixed. First, the Education variable is used in the regression analysis as though it was a continuous variable - IT IS NOT A CONTINUOUS VARIABLE - thus the regression is invalid and incorrect. The education variable needs to be included in the regression as 3 dummy variables. Second, the graphs are difficult for the reader to follow and labelling (titles and labels on axes) are insufficient

Reviewer #8: I think this is a valuable manuscript with important implications. Yours appears to be the first, or at least one of the first, studies of antimicrobial drug application to livestock in Africa. What follows are a few comments that might improve the manuscript further. My primary concerns are in bold, and I will summarize them for you here too: (1) It is unclear whether your results can be taken as representative for any geographic area or community. Probably not. If not, can the samples can be taken as “indicative,” if not statistically representative, of any larger population? This would make your results more valuable. (2) It is not clear what the three index variables (“knowledge,” “attitudes” and “practices” ) mean. As a reader, I have no idea what a high vs. low mark for “attitude” or for “practices” means, after looking at the attitude and practices questions. So at the moment, those regression results seem fairly meaningless.

See attached document for further comments.

6. PLOS authors have the option to publish the peer review history of their article (what does this mean?). If published, this will include your full peer review and any attached files.

Reviewer #1: Yes: Aurup Ratan Dhar

Reviewer #2: No

Reviewer #3: No

Reviewer #4: No

Reviewer #5: No

Reviewer #6: Yes: Andrew Barkley

Reviewer #7: No

Reviewer #8: No

---

## [Author Response · Author response to Decision Letter 0]

29 Oct 2019

Please see attached "Response to Reviewers"

---

## [Decision Letter · Decision Letter 1]

4 Dec 2019

PONE-D-19-19328R1

Towards a bottom-up understanding of antimicrobial use and resistance on the farm: A knowledge, attitudes, and practices survey across livestock systems in five African countries

PLOS ONE

Dear Dr. Caudell,

Thank you for submitting your manuscript to PLOS ONE. After careful consideration, we feel that it has merit but does not fully meet PLOS ONE’s publication criteria as it currently stands. Therefore, we invite you to submit a revised version of the manuscript that addresses the points raised during the review process.

Please review and clarify the referee's criticism again, and make your final revision.

We would appreciate receiving your revised manuscript by Jan 18 2020 11:59PM. To enhance the reproducibility of your results, we recommend that if applicable you deposit your laboratory protocols in protocols.io, where a protocol can be assigned its own identifier (DOI) such that it can be cited independently in the future. For instructions see: http://journals.plos.org/plosone/s/submission-guidelines#loc-laboratory-protocols

We look forward to receiving your revised manuscript.

Kind regards,

Arda Yildirim, Ph.D.

Academic Editor

PLOS ONE

Additional Editor Comments (if provided):

Thank you for responding to all comments and for revising the manuscript, but there are still some flaws. Please review the referee#3, 7 and 8 comments again, and make your final revision.

Reviewers' comments:

Reviewer's Responses to Questions

**Comments to the Author**

1. If the authors have adequately addressed your comments raised in a previous round of review and you feel that this manuscript is now acceptable for publication, you may indicate that here to bypass the “Comments to the Author” section, enter your conflict of interest statement in the “Confidential to Editor” section, and submit your "Accept" recommendation.

Reviewer #1: All comments have been addressed

Reviewer #2: All comments have been addressed

Reviewer #3: (No Response)

Reviewer #4: All comments have been addressed

Reviewer #7: (No Response)

Reviewer #8: All comments have been addressed

2. Is the manuscript technically sound, and do the data support the conclusions?

Reviewer #1: Yes

Reviewer #2: Yes

Reviewer #3: Partly

Reviewer #4: Yes

Reviewer #7: Partly

Reviewer #8: Yes

3. Has the statistical analysis been performed appropriately and rigorously? 

Reviewer #1: Yes

Reviewer #2: Yes

Reviewer #3: No

Reviewer #4: Yes

Reviewer #7: No

Reviewer #8: Yes

4. Have the authors made all data underlying the findings in their manuscript fully available?

Reviewer #1: Yes

Reviewer #2: Yes

Reviewer #3: Yes

Reviewer #4: Yes

Reviewer #7: Yes

Reviewer #8: Yes

5. Is the manuscript presented in an intelligible fashion and written in standard English?

Reviewer #1: Yes

Reviewer #2: Yes

Reviewer #3: Yes

Reviewer #4: Yes

Reviewer #7: Yes

Reviewer #8: Yes

6. Review Comments to the Author

Reviewer #1: The authors gave strong effort to make corrections, and it is well furnished now. So, I recommend the paper to be accepted.

Reviewer #2: (No Response)

Reviewer #3: I think that the authors have taken my comments seriously and have addressed most of them satisfactorily. I am still concerned about three points, however.

1. The first one, which is relatively minor, has to do with the definition of the education variable. Its description in Table 3 doesn’t correspond to that of three dummy variables. Please correct accordingly.

2. The second one is more substantial and has to do with the process of backward elimination of variables whose coefficients do not reach a significance level of 5%. Why that level and not, say, a 10%? Can you provide some references for the use of this econometric procedure? If collinearity is not a problem, why would you take out variables out of the model even if they are not statistically significant?

3. Finally, why would you perform the backward elimination from 3 models: one controlling for sociodemographic variables exclusively, another one including only on-farm dynamics and a final one for sources of information? My previous concern about omitted variables bias was that THE model (ONE model, not three) should include ALL the potentially important (and potentially correlated) variables.

Reviewer #4: The authors have followed my suggestions. The manuscript technically good, and the data support the conclusions. The statistical analysis has been performed appropriately and rigorously. All data underlying the findings in their manuscript fully available. The manuscript presented in an intelligible fashion and written in standard English. This manuscript can be accepted.

Reviewer #7: The 2 comments I had for the first version have (1) not been addressed or (2) the information is not provided for me to see if they addressed them. My first concern was that Education is NOT a continuous variable - yet I the table describing the Education variable they have not changed it. They have changed the regressions - so now what is in the table describing the data and the results are not consistent.

I was concerned that the graphs were difficult to follow - I find no new graphs so am unable to evaluate.

Reviewer #8: This looks much better; minor thoughts are attached for your consideration. Careful editing is still needed.

7. PLOS authors have the option to publish the peer review history of their article (what does this mean?). If published, this will include your full peer review and any attached files.

Reviewer #1: Yes: Aurup Ratan Dhar

Reviewer #2: No

Reviewer #3: No

Reviewer #4: No

Reviewer #7: No

Reviewer #8: No

---

## [Editor Report · Decision Letter 2]

27 Dec 2019

Towards a bottom-up understanding of antimicrobial use and resistance on the farm: A knowledge, attitudes, and practices survey across livestock systems in five African countries

PONE-D-19-19328R2

Dear Dr. Caudell,

We are pleased to inform you that your manuscript has been judged scientifically suitable for publication and will be formally accepted for publication once it complies with all outstanding technical requirements.

With kind regards,

Arda Yildirim, Ph.D.

Academic Editor

PLOS ONE

Additional Editor Comments (optional):

Thank you for responding to all comments and for revising the manuscript. Best wishes,
---

## [Editor Report · Acceptance letter]

2 Jan 2020

PONE-D-19-19328R2 

Towards a bottom-up understanding of antimicrobial use and resistance on the farm: A knowledge, attitudes, and practices survey across livestock systems in five African countries 

Dear Dr. Caudell:

I am pleased to inform you that your manuscript has been deemed suitable for publication in PLOS ONE. Congratulations! Your manuscript is now with our production department. 

With kind regards,

on behalf of

Dr. Arda Yildirim 

Academic Editor

PLOS ONE